# SMALLTOLARGE (S2L): Scalable Data Selection for Fine-tuning Large Language Models by Summarizing Training Loss Trajectories of Small Models

**Yu Yang**[1]    **Siddhartha Mishra**[1]    **Jeffrey Chiang**[2]    **Baharan Mirzasoleiman**[1]

[1]Department of Computer Science, [2]Department of Computational Medicine
University of California, Los Angeles (UCLA)

## Abstract

Despite the effectiveness of data selection for pretraining and instruction fine-tuning large language models (LLMs), improving data efficiency in supervised fine-tuning (SFT) for specialized domains poses significant challenges due to the complexity of fine-tuning data. To bridge this gap, we introduce an effective and scalable data selection method for SFT, SMALLTOLARGE (S2L), which trains a small model, clusters loss trajectories of the examples, and samples from these clusters to guide data selection for larger models. We prove that during fine-tuning, samples within the same loss trajectory cluster exhibit similar gradients. Then, we show that S2L subsets have a bounded gradient error w.r.t. the full data, hence guarantee convergence to the neighborhood of the optimal solution. We demonstrate through extensive experiments that S2L significantly improves data efficiency in SFT for mathematical problem-solving, reducing the training data requirement to just 11% of the original MathInstruct dataset [64] to match full dataset performance while outperforming state-of-the-art data selection algorithms by an average of 4.7% across 6 in- and out-domain evaluation datasets. Remarkably, selecting only 50K data for SFT, S2L achieves a 32.7% accuracy on the challenging MATH [19] benchmark, improving Phi-2 [28] by 16.6%. In clinical text summarization on the MIMIC-III dataset [21], S2L again outperforms training on the full dataset using only 50% of the data. Notably, S2L can perform scalable data selection using a reference model $100\times$ smaller than the target model, proportionally reducing the computational cost. [1]

## 1  Introduction

In recent years, large language models (LLMs) have revolutionized artificial intelligence by demonstrating an unprecedented ability to understand and generate human language [7]. Among all the contributing factors, the quality and selection of data is becoming increasingly recognized for its importance in training LLMs effectively. Recent research indicates that LLMs benefit more from training for additional epochs on carefully curated data rather than on larger, uncurated ones during pretraining [48] and instruction fine-tuning [67], making data selection one of the most promising means to unlock the next level of LLMs' language capability. However, while generalist models obtained through pre-training or instruction fine-tuning excel in *general language tasks*, they may not deliver optimal outcomes in *specialized domain*, such as mathematics [3, 31, 63, 30, 64], code [42, 32], medicine [43, 44, 9], or finance [57, 9]. These domains are not only critical for real-world applications but also hold substantial economic and societal impacts.

---

[1]Code is available at https://github.com/BigML-CS-UCLA/S2L.

38th Conference on Neural Information Processing Systems (NeurIPS 2024).

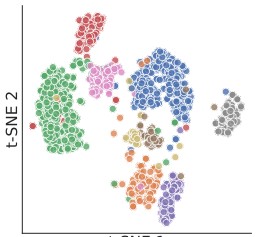
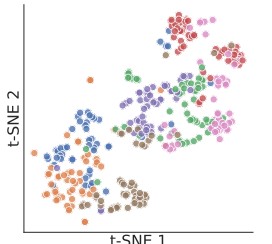
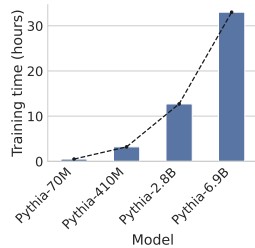

(a) Hidden states of the Pile on pretrained Pythia-410M

(b) Hidden states of MathInstruct on pretrained Pythia-410M

(c) Increase in training time as the size of the model scales up

Figure 1: Existing data selection methods depend heavily on the feature representations from a reference model, which makes their effectiveness vulnerable to the quality of training on the target domain [34]. For supervised fine-tuning (SFT), while pretrained models can effectively separate topics (shown in different colors) in natural language (Figure 1a), they struggle with fine-tuning data that deviates from the pretraining distribution (Figure 1b). Additionally, the cost of training a reference model escalates with model size (Figure 1c), making existing data selection methods for large models prohibitively expensive.

To maximize performance in specialized domains, models fine-tuned on domain data offer superior capabilities over generalist models [20]. Yet, maximizing the data efficiency in supervised fine-tuning (SFT) for specialized domains remains a challenging and under-explored problem. Firstly, heuristic approaches that are effective in the instruction fine-tuning stage, like manual curation [67] or using advanced models such as GPT-4 for dataset evaluation [8], are less reliable due to the need for specialized knowledge and become costly with large volumes of uncurated fine-tuning data. Beyond these heuristic methods, other approaches rely on generating representations for each training example using a reference model, often utilizing metrics like perplexity [34], confidence [46, 53], or hidden states [1, 48, 61, 4] as features. However, these techniques also fall short in SFT for specialized domains for two reasons: (1) the significant shift between pretraining and SFT data can render these metrics less informative (Figure 1b), and (2) the computation and memory demands associated with generating representations for each training example become prohibitive, as these specialized domains often require larger models, some with up to 540 billion parameters [10, 43], leading to substantial scalability challenges (Figure 1c).

To tackle the challenges of data efficiency in SFT for specialized domains, we present SMALLTOLARGE (S2L), an effective and scalable data selection algorithm. S2L operates by first gathering training loss trajectories for each training example using a small model. These trajectories are then clustered, and similar number of examples are selected from these clusters uniformly at random. This process is grounded in our theoretical findings that examples within the same cluster exhibit similar gradients during training, thereby affecting the model similarly. Consequently, subsets sampled from these clusters have a bounded gradient error w.r.t. the full data, allowing for training a comparable model with only a subset of data. Furthermore, we provide a convergence rate analysis for training on these subsets, establishing a robust theoretical foundation for S2L's effectiveness and efficiency.

To validate S2L's effectiveness, we applied it to the challenging tasks of SFT for (1) mathematical problem-solving and (2) clinical text summarization. Our experiments on MathInstruct [64] shows that S2L can significantly reduce the required training data size to just 11% of the original dataset size while still matching the performance levels of the full dataset, outperforming current state-of-the-art one-shot and online data selection algorithms by an average of 4.7% across 6 in- and out-domain evaluation datasets. Remarkably, on the MATH benchmark [19], S2L attained a 32.7% accuracy with just 50K data points, improving the best open-sourced model under 3 billion parameters, Phi-2, by 16.6%. For clinical text summarization tasks on the MIMIC-III [21] dataset, S2L outperforms training on the full dataset, using only half of the data. Unlike existing methods that require training and getting features from large models, S2L achieves superior data efficiency using a model with as few as 70 million parameters, which is $100\times$ smaller than the largest target model we train with 7 billion parameters.

## 2 Related Work

**Foundations of Data Selection.** Data selection has been well studied for small models and classification tasks. There are one-shot algorithms that select data based on rankings of the proposed training statistics, for example, the L2-norms of error and gradient vectors (EL2N and GraNd) [39], confidence and its variability across epochs [46], and the number of times each example is learned but then forgot at the subsequent training step [49]. Besides these heuristic indicators, there are embedding-based pruning algorithms [45] and online selection algorithms with theoretical performance guarantees for efficiency [35, 23, 24, 40, 60] and robustness [59, 62, 16]. Coleman et al. proposed to use the intermediate feature representation of a small proxy model to select data for image classification. Most recently, data selection has shown great potential in more substantial training speedup when implemented on near-storage hardware [41], and data selection beyond supervised learning of image data, e.g., for self-supervised learning [22] and multimodal learning [1, 33], also emerged.

**Data Efficient Training of Large Language Models.** For the pre-training of LLMs, Marion et al. studied data quality indicators including Perplexity, Error L2-Norm (EL2N) [39], and memorization ranking [5], and found training on examples with middle Perplexity rankings outperforms training on examples selected based on the other two metrics, and sometimes even outperforms training on the entire dataset. Tirumala et al. uses pre-trained model embeddings to select data for LLM pre-training. The proposed algorithm, D4, first applies an embedding-based data de-duplication algorithm [1] and then discards data points that are the closest to the K-Means cluster centroids in the embedding space [45] to ensure diversity. On fine-tuning LLMs, existing work on data efficiency primarily focused on manually curating high-quality instructions [67], or using strong closed-source models (e.g., GPT-4 [2] or ChatGPT) to rate the quality of each training example [18, 27, 8]. Bhatt et al. implemented an experimental design framework to evaluate the existing data selection methods for instruction fine-tuning of LLMs and found selecting facility locations based on hidden representations (i.e., embeddings) is the most effective. As the only data selection algorithm for specialized domains, SCIP [61] focuses on pruning low-quality code data for training code LLMs. Since it relies on breaking the code syntax to understand the characteristics of low-quality code in the embedding (i.e, hidden states) space, adapting SCIP to domains other than Python code data is non-trivial.

**Adapting Large Language Models for Specialized Domains.** The rapid development of large language models (LLMs) gives rise to new state-of-the-art models in specialized domains. For mathematical reasoning, Galactica [47], MINERVA [26] and Llemma [3] continue to train an LLM on large-scale math-related web data to improve a model's general scientific reasoning; WizardMath [31] and TinyGSM [30] fine-tune LLMs using supervised data. Similarly for medical LLMs, Cheng et al. continued training pre-trained LLMs on medical text, and [43, 44] fine-tuned PaLM with instruction prompt tuning on medical domain data.

## 3 Problem Formulation

**LLM Fine-tuning Objective.** Consider a transformer-based language model, parameterized by $\boldsymbol{\theta}$, and denoted as $p_{\boldsymbol{\theta}}$. This model, when provided with a sequence of prompt tokens $\mathbf{x} = (x_1, \ldots, x_M)$, generates a sequence of response tokens $\mathbf{y} = (y_1, \ldots, y_L)$. The conditional probability of generating $\mathbf{y}$ given $\mathbf{x}$ is then formulated as

$$p_{\boldsymbol{\theta}}(\mathbf{y}|\mathbf{x}) = \prod_{l=1}^{L} p_{\boldsymbol{\theta}}(y_l|\mathbf{y}_{1:l-1}, \mathbf{x}). \tag{1}$$

Note that $\mathbf{y}_{1:0}$ is an empty sequence. To adapt the pre-trained LLM for a specialized domain of distribution $\mathcal{D}$, supervised fine-tuning (SFT) is usually employed with a domain-specific training dataset $D_{\text{train}} = \{(\mathbf{x}, \mathbf{y})_i\}_{i=1}^{n} \sim \mathcal{D}$ containing pairs of prompt $\mathbf{x}$ and annotated response $\mathbf{y}$. The fine-tuning objective is thus to minimize the following negative log likelihood loss, expressed as:

$$\min_{\boldsymbol{\theta}} \mathcal{L}(\boldsymbol{\theta}, D_{\text{train}}) = -\frac{1}{n} \sum_{(\mathbf{x}, \mathbf{y})_i \in D_{\text{train}}} \big[ \log p_{\boldsymbol{\theta}}(\mathbf{y}_i|\mathbf{x}_i) \big]. \tag{2}$$

**Data Selection Objective.** In a general setting for data selection, we consider a target language model $p_{\boldsymbol{\theta}}$ with parameters $\boldsymbol{\theta}$. Given a fixed data budget $B$, which constrains the number of data points

that can be used for training, our objective is to select a subset $S \subseteq D_{\text{train}}$ to train the target model, such that it obtains a superior generalization performance. In practice, the subset $S$ is selected based on a reference model $r_\phi$ parameterized by $\phi$, which generates representations, confidence scores, or other metrics for each data point $(\mathbf{x}, \mathbf{y})_i \in D_{\text{train}}$, denoted by $r_\phi((\mathbf{x}, \mathbf{y})_i)$, which will be utilized by a data selection algorithm to produce $S$.

In existing data selection algorithms, $\phi$ is commonly either weights of the pre-trained target model or a target model that has been fully trained on the dataset $D_{\text{train}}$. However, as evidenced by Figure 1, representations generated by the pretrained model may not always be good enough for data selection in specialized domains, and fine-tuning the target model significantly increases the computational cost of data selection.

## 4 Methodology

Training a large target model to obtain feature representations for each example in $D_{\text{train}}$ can be computationally intensive. However, a recent finding demonstrates that the training dynamics of most examples are consistent across differently sized models of the same family, and this phenomena even generalizes across different model families [58]. Our proposed method, **SMALLTOLARGE (S2L)**, leverages *loss trajectories* of training examples collected during fine-tuning a *small* reference model on the full or a subset of training data.

**Loss Trajectory.** Let $\phi^{(t)}$ be the parameters of a small LM during training on $D_{\text{train}}$ at times $t_q, q \in \{1, ..., T\}$. S2L records the loss trajectory for each data point $i$ at times $t_q$ during training the reference model $[\mathcal{L}_i^{\text{proxy}}(\phi^{(t_1)}), \ldots, \mathcal{L}_i^{\text{proxy}}(\phi^{(t_T)})]$ where

$$\mathcal{L}_i^{\text{proxy}}(\phi^{(t)}) = \mathcal{L}^{\text{proxy}}(\phi^{(t)}, (\mathbf{x}_i, \mathbf{y}_i)) = -\log p_{\phi^{(t)}}(\mathbf{y}_i | \mathbf{x}_i), \tag{3}$$

and $T$ is the length of the loss trajectory. Note that $\phi^{(t)}$ is trained for a fixed number of iterations from $\phi^{(t-1)}$.

Assume the parameter vector $\boldsymbol{\theta}^{(t)}$ represents the parameters of the target model at the time $t$. Define $\mathbf{L}_i^{\text{proxy}} = [\mathcal{L}_i^{\text{proxy}}(\phi^{(t_1)}), \ldots, \mathcal{L}_i^{\text{proxy}}(\phi^{(t_T)})]$ and $\mathbf{L}_i^{\text{target}} = [\mathcal{L}_i^{\text{target}}(\boldsymbol{\theta}^{(t_1)}), \ldots, \mathcal{L}_i^{\text{target}}(\boldsymbol{\theta}^{(t_T)})]$ as the training loss trajectory of the example $i$ on the small proxy model and the large target model, respectively. Let $\boldsymbol{H}_i \in \mathbb{R}^{d \times d}$ be the Hessian matrix for each example $i$ and assume that the loss function for each example during fine-tuning can be modeled by a second-order Taylor approximation with bounded curvature ($c \leq \|\boldsymbol{H}_i\| \leq C$), a reasonable assumption in fine-tuning settings. The following lemma shows that examples with similar loss trajectories on the proxy model have similar gradients throughout the training of the target model.

**Theorem 4.1.** *If examples $i$ and $j$ have similar loss trajectories on the proxy model, i.e., $\|\mathbf{L}_i^{proxy} - \mathbf{L}_j^{proxy}\| \leq \epsilon$, and their loss trajectories on the proxy and target model is similar, i.e., $\|\mathbf{L}_p^{proxy} - \mathbf{L}_p^{target}\| \leq \delta$ for $p \in \{i, j\}$, then $i$ and $j$ have similar gradients throughout training the target model:*

$$\|\nabla \mathcal{L}_i^{target}(\boldsymbol{\theta}) - \nabla \mathcal{L}_j^{target}(\boldsymbol{\theta})\| \leq \frac{2\epsilon' + 2CD^2}{d} = \Delta. \tag{4}$$

*where $\epsilon' = \epsilon + 2\delta$ and $\|\boldsymbol{\theta}\| \leq D$ for all $t$.*

The proof of Theorem 4.1 can be found in Appendix A.1. Theorem 4.1 shows that examples with similar loss trajectories have similar gradients during the training, thereby influencing the model in a similar manner.

**Data selection from Loss Trajectory Clusters.** Once the loss trajectories are recorded on the proxy model, we apply a clustering algorithm to group examples based on the similarity of their loss trajectories. This results in a set of clusters $\{C_1, C_2, \ldots, C_K\}$, where each cluster $C_i$ contains examples with similar loss and gradient trajectory throughout the training:

$$C_i = \{(\mathbf{x}, \mathbf{y})_j \in D_{\text{train}} | i = \arg \min_{j \in [K]} d(\mathbf{L}_j, \mathbf{L}_{\bar{C}_j})\}, \tag{5}$$

where $\mathbf{L}_{\bar{C}_i}$ is the centroid of the loss trajectories in cluster $C_i$, and $d(\cdot, \cdot)$ is a distance metric, such as Euclidean distance, used for clustering. For datasets that contain different sources of data, we cluster each source separately.

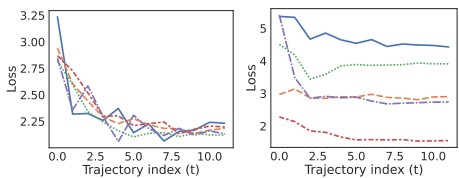

(a) In the same cluster.  (b) In different clusters.

Figure 2: Examples in the same clusters have very similar loss trajectories (Figure 2a) while the loss trajectories of examples in different clusters are very different (Figure 2b).

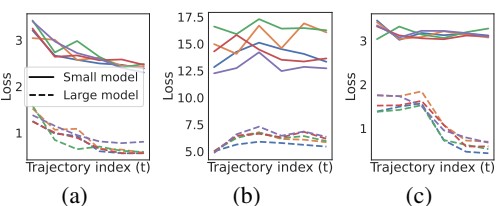

(a)  (b)  (c)

Figure 3: Examples in the same clusters of training trajectories on a small model (Pythia-70M) also have similar training trajectories on a large model (Pythia-2.8B), even if the trends may not be the same on both models.

---

**Algorithm 1** Data Selection Based on Training Trajectories (S2L)

---

**Require:** Training dataset $D_{\text{train}}$ with corresponding training trajectories, a fixed data budget $B$, number of clusters $K$.
**Ensure:** Subset $S \subseteq D_{\text{train}}, |S| \leq B$.
 1: Initialize $S$ as an empty set.
 2: Train a small proxy model and cluster examples in (each data source of) $D_{\text{train}}$ based on their loss trajectories and sort them by size to get $\mathcal{C} = \{C_1, C_2, \ldots, C_K\}$.
 3: **for** each cluster $C_k$ in $\mathcal{C}$ **do**
 4:   Calculate $R_k$, the number of examples to randomly sample from $C_k$, i.e., $R_k = (B - |S|)/(K - k + 1)$.
 5:   **if** $|C_k| \leq R_k$ **then**
 6:     $S \leftarrow \{S \bigcup C_k\}$.
 7:   **else**
 8:     $S \leftarrow \{S \bigcup S_k\}$, where $S_k \subset C_k$ is selected uniformly at random from $C_k$ and $|S_k| = R_k$
 9:   **end if**
10: **end for**
11: Return $S$

---

As shown in Figure 2, clustering algorithms can effectively find groups of examples with similar training dynamics. In Figure 3, we empirically show that we can identify groups of examples with similar training dynamics on a larger model by clustering the training trajectories of $D_{\text{train}}$ on a smaller proxy model. With the clusters formed, the data selection strategy selects equal number of examples at random from all clusters, as detailed in Algorithm 1. In doing so, it effectively prioritizes selecting examples from smaller clusters. This is particularly important for datasets containing multiple imbalanced sources. In this setting, training and test distributions often differ, and balanced selection from clusters ensures superior test performance on all groups in the test data.

The following theorem shows that, under the assumptions of Theorem 4.1, training with Incremental Gradient (IG) methods on the subset selected by S2L converges to a close neighborhood of the optimal solution found by training the target model on the full dataset. IG methods such as Stochastic Gradient Descent (SGD) update parameters iteratively based on the gradient of the loss of individual examples, multiplied by stepsize $\alpha$. Formally,

$$\boldsymbol{\theta}^{t+1} = \boldsymbol{\theta}^t - \alpha\nabla\mathcal{L}_i^{\text{target}}(\boldsymbol{\theta}^t). \tag{6}$$

**Corollary 4.2.** *Under the assumptions of Theorem 4.1, applying IG with stepsize $\alpha$ to subsets found by S2L, converges to the neighborhood of the optimal solution, as follows:*

$$\|\boldsymbol{\theta}^{t+1} - \boldsymbol{\theta}^*\|^2 \leq (1 - \alpha c)^{t+1}\|\boldsymbol{\theta}^t - \boldsymbol{\theta}^*\|^2 + 2\xi R/c^2 + \alpha B^2(r_{\min}/k)^2 \boldsymbol{g}_{\max}^2 \tag{7}$$

*where $c \leq \|\boldsymbol{H}\|$, $B = k \cdot K$ is the total size of the subset, $\boldsymbol{g}_{\max}$ is the largest gradient norm of individual examples during training, $r_{\min} = \min_j |C_j|, r_{\max} = \max_j |C_j|$, $R = \min\{d_0, B\boldsymbol{g}_{\max} + \xi/c\}$ and $d_0 = \|\boldsymbol{\theta}^0 - \boldsymbol{\theta}^*\|$ is the initial distance to the optimal solution $\boldsymbol{\theta}^*$, and $\xi$ is given by:*

$$\xi = K[r_{\min}\Delta + (r_{\max} - r_{\min})\boldsymbol{g}_{\max}]. \tag{8}$$

The proof can be found in Appendix A.2.

## 5  Experiments

In this section, we present the comprehensive experiments conducted to evaluate the efficacy of the proposed data selection method, S̲MALLTO̲LARGE (S2L), across two challenging domains (mathematical reasoning and clinical text summarization).

### 5.1 Baselines

We systematically compare S2L against a comprehensive set of open-sourced data selection methods. These methods are categorized based on the type of representation they use and selected as the most representative or best-performing methods as identified in prior work. These include: (1) **Random Sampling**; selecting examples with the (2) **Least Confidence** [4] or (3)**Middle Perplexity** [34]; (4) **High Learnability**, determined by the loss decrease before and after full fine-tuning [68]; and (5) **Facility Locations** selection based on hidden states [4]. Additionally, we incorporate one online selection techniques: (6) **Confidence Curriculum** proposed by Varshney et al., which selects examples with decreasing confidence during the training. Given that the optimal reference model may vary for each one-shot selection method, we ensure a fair comparison by adopting the approach used in [34], which runs each method with both the fully fine-tuned target model and an early fine-tuning checkpoint as the reference model. We report the best results from these setups.

### 5.2 Specialized Domain 1: Mathematical Reasoning

**Training Settings.** We focus on fine-tuning using the **MathInstruct** dataset [64] with 262,040 training examples for its comprehensive coverage of diverse mathematical fields and its capability in training models to achieve state-of-the-art performance on the standard evaluation benchmarks. We employ the open-source model suites Pythia [6], Phi-2 [28], Llama-2 [50] as our base models to validate our S2L algorithm and directly compare its performance against the state-of-the-art.

**Evaluation Datasets.** We follow the framework established in [64] for a comprehensive assessment using several well-regarded datasets, including in-domain and out-of-domain datasets. For the in-domain datasets, we consider **GSM8K** [11], **MATH** [19], and **NumGLUE** [36]. For the out-of-domain datasets, we consider **SVAMP** [38], **Mathematics** [13], **SimulEq** [25]. These datasets collectively span a diverse range of mathematical subjects, such as algebra, probability, number theory, calculus, and geometry. Additionally, some questions in these datasets require the application of commonsense, reading comprehension, and multi-step reasoning. All questions are open-formed.

**Evaluation Metric.** We use the standard evaluation metric for open-formed questions, **exact match**, which measures the model's accuracy by comparing its generated answers against the correct solutions. For an answer to be considered correct, it must match the reference solution precisely.

More details about the settings and baseline implementations can be found in Appendix B.

#### 5.2.1 Setting 1: Less Data for Better Models

In the first setting, we standardize the number of training steps to correspond to 3 epochs on the full dataset, aligning with [64]. This allows us to maintain a consistent training schedule across different methods and data budgets, ensuring fair and accurate comparisons of the quality of data.

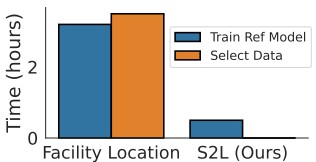

**SCALING THE DATA: SOTA algorithms fail with small data budgets while S2L stands out across data scales.** In Figure 4, we compare S2L against the baselines from Section 5.1 on Pythia-410M across varying data sizes. The training trajectories used by S2L are based on Pythia-70M, a model approximately 6x smaller than Pythia-410M. With the same number of training steps as the full training, S2L exceeds the full dataset's performance using only 30K examples, only 11% of the full dataset. It leads the runner-up baselines by an average of 4.7%, 4.6% and 2.4% with data budget 30K, 50K and 100K across all six evaluation datasets. While state-of-the-art data selection algorithms

Figure 5: Wall-clock time required to train the reference model and select 100K data from MathInstruct for training Pythia-410M.

like Facility Locations [4] and High Learnability [68] have decent performance with a large enough data budget (i.e., 100K), all SOTA algorithms except S2L cannot even outperform the random sampling baseline when the allowed data size is small (i.e., 30K). Unlike the existing algorithms, S2L consistently outperforms all baselines and even full training across all data sizes. Note that compared to the runner-up algorithm in this setting, Facility Locations, the cost of S2L is much lower in both training the reference model and data selection stages (Figure 5), and therefore more scalable to both larger target models or larger data sizes.

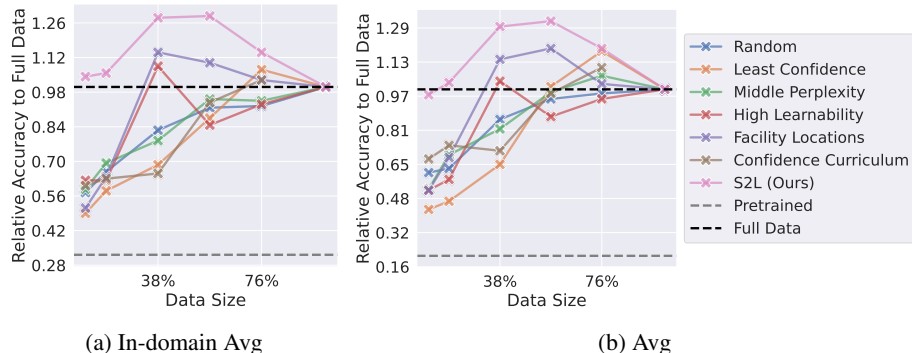

(a) In-domain Avg             (b) Avg

Figure 4: **Data Scaling:** Accuracies (↑) on in-domain and out-of-domain datasets using Pythia-410M. Data size refers to the total number of unique training examples used. All experiments in this table share the same total training steps and learning rate schedule (see Section 5.2). See breakdowns in Figure 14.

Table 1: **Less Data, Same Compute:** Zero-shot accuracies (%, ↑) when S2L and the baselines select 50K data to train with the same number of iterations as the full-data training. Results surpassing full training are highlighted in bold. Figure 4 follows the same setting but uses the Pythia-410M model.

| TARGET MODEL | FINE-TUNING DATA | IN-DOMAIN | | | | OUT-DOMAIN | | | |
|---|---|---|---|---|---|---|---|---|---|
| | | GSM8K | MATH | NUMGLUE | AVG | SVAMP | MATHEMATICS | SIMULEQ | AVG |
| PHI-2 (2.7B) | (PRETRAINED) | 53.4 | 16.1 | 34.9 | 34.8 | 67.9 | 31.1 | 27.4 | 38.5 |
| | RANDOM | 67.9 | 30.1 | 60.7 | 52.9 | 77.1 | 51.2 | 37.5 | 54.1 |
| | HIGH LEARNABILITY | 59.4 | 25.2 | 62.1 | 48.9 | 76.6 | 41.8 | 27.2 | 48.7 |
| | MIDDLE PERPLEXITY | 66.4 | 29.5 | 54.1 | 50.0 | 74.8 | 50.4 | 39.8 | 52.5 |
| | LEAST CONFIDENCE | 61.7 | 24.7 | 67.0 | 51.1 | 76.5 | 43.3 | 52.5 | 54.3 |
| | FACILITY LOCATIONS | 66.2 | 31.3 | 62.4 | 53.3 | 74.4 | 58.4 | 34.6 | 54.5 |
| | **S2L(OURS)** | **69.1** | **32.6** | **65.7** | **55.8** | **79.6** | 56.4 | 40.1 | 57.3 |
| | FULL-262K | 68.3 | 32.6 | 64.3 | 55.1 | 78.4 | 58.4 | 44.2 | **57.7** |

SCALING THE MODEL: **Data selected by S2L can transfer to larger models in different model suites.** We also test whether this subset, chosen using Pythia-70M, can effectively train larger models beyond 410M and models outside the Pythia suite. As shown in Table 1, our experiments with Phi-2 reveal that fine-tuning on only 50K S2L-selected data again outperforms full dataset training on the most challenging MATH [19] benchmark improving the pretrained Phi-2 by $16.6\%$ and is more data efficient than training on the full MathInstruct dataset to get the same performance.

### 5.2.2 Setting 2: Less Data for Faster Training

The second setting we consider is when fixing the number of times each example can be seen over the entire course of training, directly translating smaller datasets into reduced training and storage costs. This is particularly beneficial for large models that would require extensive training times without data selection. By experimenting with models of larger sizes than the previous setting, we observe in Table 2 that S2L can achieve comparable performance to full-data training when using only 50% data and thereby reducing both the data storage space and the training time by half.

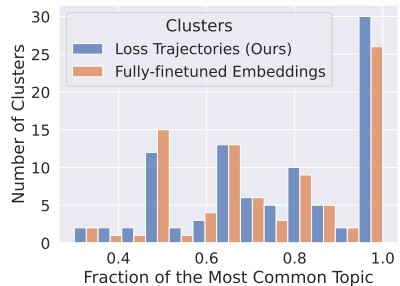

Figure 6: Distribution of the coverage of top-1 topic in each cluster. Taller bars on the right end of the plot indicate clusters with a higher concentration of a single topic and therefore suggest a grouping of similar examples.

### 5.2.3 Why is S2L So Effective?

**Examples in Clusters Encode the Same Knowledge/Skill.** In Appendix C, we compare actual training examples in MathInstruct that get clustered together due to their similar training trajectories on the small Pythia-70M model. We observe that examples in the same cluster are of the same type and related to the same knowledge/skill, e.g., open-formed algebra questions (Figure 15), examples requiring extracting useful information from long text and writing programs

Table 2: **Less Data, Same Epochs:** Zero-shot accuracies (%, ↑) when S2L trains 50% data for the same number of epochs as the full-data training. S2L can achieve comparable performance to full-data training while reducing both the data storage space and the training time by half.

| TARGET MODEL | FINE-TUNING DATA | IN-DOMAIN | | | | OUT-DOMAIN | | | |
|---|---|---|---|---|---|---|---|---|---|
| | | GSM8K | MATH | NUMGLUE | AVG | SVAMP | MATHEMATICS | SIMULEQ | AVG |
| PHI-3-MINI (3.8B) | (PRETRAINED) | 74.5 | 26.5 | 52.1 | 51.1 | 83.7 | 44.3 | 34.8 | 52.7 |
| | **S2L-50%(OURS)** | 76.3 | 42.5 | 76.4 | 65.1 | 83.8 | 62.1 | 51.6 | 65.4 |
| | FULL | 76.4 | 42.9 | 75.3 | 64.9 | 84.6 | 60.2 | 51.9 | 65.2 |
| LLAMA-2-7B | (PRETRAINED) | 3.1 | 4.2 | 16.5 | 7.9 | 14.1 | 8.3 | 2.3 | 8.1 |
| | **S2L-50%(OURS)** | 53.3 | 28.9 | 65.0 | 49.1 | 65.1 | 45.2 | 31.9 | 48.2 |
| | FULL-262K [64] | 52.2 | 30.4 | 60.5 | 47.7 | 65.3 | 43.9 | 50.2 | 50.4 |

(Figure 16), and multiple choice questions that require multi-step reasoning (Figure 17), etc. Therefore, by sampling from different clusters, we make sure the selected examples cover the knowledge required for all topics and skills required for all types of questions.

**Loss Trajectories can Capture the Similarity Between Data Points As Much As Embeddings of a Fully Fine-tuned Model.** We conducted a quantitative analysis to assess how effectively S2L identifies similar examples using loss trajectories from a small model. Assuming math problems under the same topic require similar knowledge and share question formats, we used unknown topic labels during S2L's data selection to check if each cluster predominantly contains a single topic. By calculating the fraction of the most common topic in each cluster and plotting this in Figure 6 (with K=100, excluding clusters of size one), we compared the loss trajectory clusters from S2L (in blue) against those from the embeddings of a fully fine-tuned Phi-2 model (in orange)—considered the ground truth for similarity. Results show that most clusters formed by S2L using the Pythia-70M model are based on a single topic and capture topic similarities more effectively than those from the Phi-2 model's embeddings. This analysis not only confirms the homogeneity within S2L clusters but also highlights the computational efficiency of using loss trajectories on small models to identify representative examples.

## 5.3 Specialized Domain 2: Clinical Text Summarization

S2L can improve data efficiency not only for fine-tuning data not only in mathematics but also in other specialized domains. This subsection explores its application to clinical text summarization within radiology reports. This task involves processing the detailed analysis and results listed in the findings section of a radiology report and distilling them into a concise impression section. Such summaries are crucial for providing clinicians with quick and actionable insights from radiological studies.

**Dataset & Setup.** We use the **MIMIC-III** dataset [21], a comprehensive collection of radiology reports and findings authored by attending physicians in routine clinical care. We use the same preprocessing procedures as [14, 15] to extract the findings and impression sections and remove invalid reports. Given that access to MIMIC-III requires specific credentials, we provide a synthetic example of a radiology report generated by GPT-4 [2] for illustrative purposes in Table 3. We employ the Pythia-1B model and keep the training setting consistent with the mathematical reasoning task.

**Evaluation.** Our evaluation of generated clinical summaries on the MIMIC-III dataset's test split employs three key metrics as recommended in [52, 51]: (1) **BLEU** [37], which measures word sequence overlap between the generated and reference texts; (2) **ROUGE-L** [29], assessing the longest common word sequence; and (3) **BERTScore** [65], evaluating semantic similarity using BERT's contextual embeddings. These metrics together offer a comprehensive evaluation, ensuring our summaries are not only precise in language but also meaningful and coherent in the context of clinical information. We compare S2L to random selection, a surprisingly strong baseline as evidenced in Section 5.2, to check the validity of the data selection problem on this dataset and then compare it to training on the full dataset to assess its effectiveness.

**Results.** We compare using 30K examples selected by random vs. selected through S2L. Even with only half of the data, the model trained with S2L selected data achieves similar BLEU and significantly higher ROUGE-L and BERTSCore compared to the model trained on the entire 61.5K data. Meanwhile, training on randomly selected 30K examples performs worse than training on the full dataset on all 3 metrics. These results together demonstrate S2L's effectiveness.

## 5.4 Ablation Studies

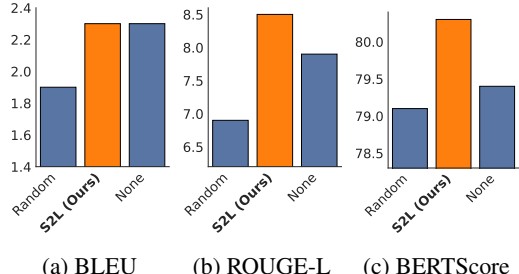

|  (a) BLEU  |  (b) ROUGE-L  |  (c) BERTScore  |

Figure 7: Performance (↑) of models trained on (1) **random**ly selected 30K examples, (2) **S2L** selected 30K examples, and (3) full 61K examples (**none**) evaluated with 3 different metrics. The minimum value on the y-axis is the performance of the model before fine-tuning. S2L improves the data efficiency for the clinical text summarization task by outperforming training on the full dataset with only less than half of the data.

We conduct ablation studies on MathInstruct and Pythia-410M to further understand the best practices for using S2L.

**S2L is robust w.r.t. the length of the trajectories but can benefit more from longer trajectories.** Figure 8 compares models trained with data selected by S2L based on training trajectories of different lengths. The shorter trajectories are derived from a uniform sample of the longer trajectories. From the small slopes of the lines, we can conclude that S2L is relatively robust to the length of the training trajectories. Nevertheless, we can also observe a slight increase in the performance on some of the datasets when longer trajectories are used, so having longer trajectories is still preferred.

**S2L can utilize training trajectories collected at any stage of training but preferably denser ones.** With the length of the trajectories fixed to 4, we can observe in Figure 9 that denser trajectories recorded at any training stage (early, middle, or late) are more helpful for S2L than trajectories recorded sparsely throughout the training.

**S2L does not require the full training data to train the proxy and can scale efficiently to larger datasets.** To further demonstrate the scalability of the proposed S2L method, we conducted experiments by training the proxy on a smaller sample of the data (100K examples) for the same number of epochs (3 epochs) and saving the loss for all examples. The results, shown in Figure 10, confirm that S2L remains effective when the proxy model is trained on a smaller subset of training data and therefore is scalable to larger datasets without a proportional increase in computational costs.

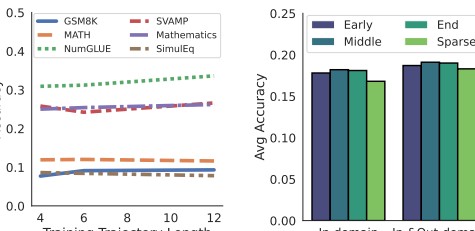

Figure 8: S2L is robust to the length of training trajectories.

Figure 9: S2L prefers dense trajectories over sparse ones.

**S2L is robust across different clustering parameter values for K.** We conducted detailed experiments varying the clustering parameter K, as shown in Figure 11. The results demonstrate that S2L maintains high performance across different values of K, highlighting the robustness of our method to different clustering parameter choices. We chose K=100 for our experiments as it provided the best average accuracy across the evaluation datasets for the math reasoning task.

**S2L remains effective and efficient compared to using full data when trained for the same number of epochs.** Figure 12 illustrates the relative accuracy to full data across different epochs, comparing S2L-selected data and full data with the same number of epochs. Both in-domain and overall average accuracy are shown. S2L demonstrates superior performance with fewer data and fewer training iterations.

**S2L supports a range of small models as effective proxies.** To understand whether different small models could serve as effective proxies, we used GPT-2 (124M) and Pythia-160M as proxy models for data selection to train Pythia-410M. The results, illustrated in Figure 13, show that both proxy models perform comparably in guiding the data selection, demonstrating the versatility and effectiveness of using different small models for S2L.

# 6 Conclusion and Limitations

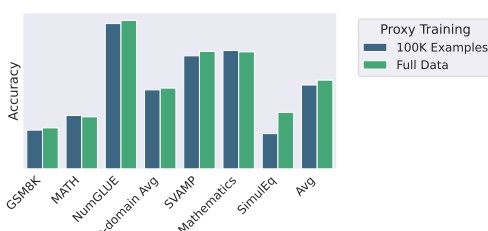

Figure 10: Per-dataset and average accuracy comparing proxy training on 100K examples and full data. S2L remains effective.

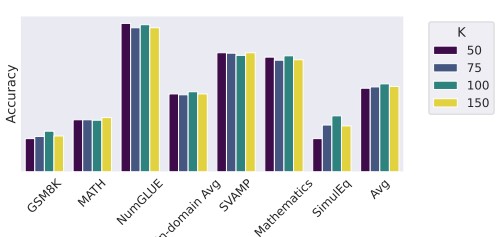

Figure 11: Per-dataset and average accuracy across different values of the clustering parameter $K$. S2L is relatively robust to the choice of $K$.

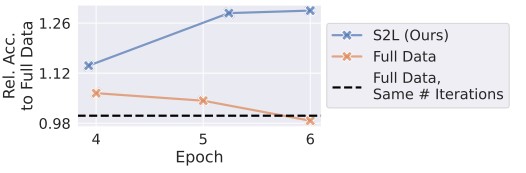
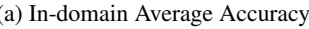

(a) In-domain Average Accuracy

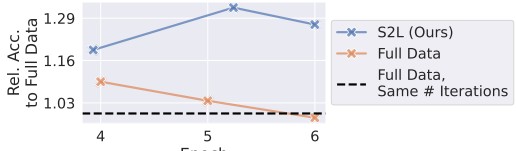

(b) Overall Average Accuracy

Figure 12: Relative accuracy to full data across different epochs, comparing S2L-selected data and full data. S2L achieves superior performance with fewer data and fewer training iterations.

In this work, we introduced SMALLTOLARGE (S2L), a scalable data selection method to improve the data efficiency of supervised fine-tuning (SFT) for large language models (LLMs) in specialized domains. By clustering data points based on their training dynamics on smaller models and balanced sampling from all clusters, S2L significantly reduces the required training data size without compromising performance compared to using the entire training dataset. Our comprehensive experiments across the mathematical problem-solving and clinical text summarization domains demonstrate the effectiveness of S2L.

Our study does come with its limitations. S2L has been only tested within two domains, mathematics and medicine, and on models up to 7 billion parameters, constrained by our computational resources. Additionally, our experiments employed a fixed training schedule across all methods without further optimization or hyperparameter tuning for each method, including S2L. This unified approach, while it ensures a fair comparison, may not fully capture the potential performance improvement that could be achieved with more tailored training strategies. We encourage further research to extend the application of S2L across a broader spectrum of domains and investigate the impact of hyperparameter tuning on its effectiveness.

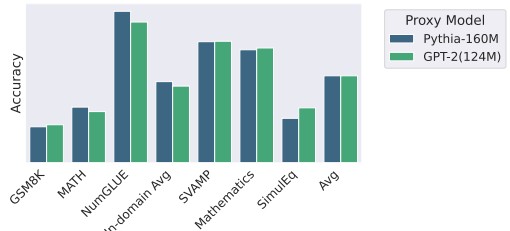

Figure 13: Per-dataset and average accuracy comparison between using different proxy models (Pythia-160M and GPT-2 (124M)) for data selection. Using both proxy models show comparable performance, demonstrating the effectiveness of different small models as reference models for S2L.

# Acknowledgments

This research was partially supported by the National Science Foundation CAREER Award 2146492, National Science Foundation 2421782 and Simons Foundation, Cisco Systems, Optum AI, and a UCLA Hellman Fellowship.

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

# A Proofs

## A.1 Proof of Theorem 4.1

*Proof.* From the assumption that the loss trajectories of examples on the proxy and target models are close:

$$\|\mathbf{L}_i^{\text{proxy}} - \mathbf{L}_i^{\text{target}}\| \leq \delta, \quad \forall i. \tag{9}$$

Since $i$ and $j$ are in the same cluster $C_k$ based on the proxy model, we have:

$$\|\mathbf{L}_i^{\text{proxy}} - \mathbf{L}_j^{\text{proxy}}\| \leq \epsilon. \tag{10}$$

Using the triangle inequality:

$$\|\mathbf{L}_i^{\text{target}} - \mathbf{L}_j^{\text{target}}\| \leq \|\mathbf{L}_i^{\text{target}} - \mathbf{L}_i^{\text{proxy}}\| + \|\mathbf{L}_i^{\text{proxy}} - \mathbf{L}_j^{\text{proxy}}\| + \|\mathbf{L}_j^{\text{proxy}} - \mathbf{L}_j^{\text{target}}\| \leq 2\delta + \epsilon = \epsilon'. \tag{11}$$

Therefore, at any iteration $t$:

$$|\mathcal{L}_i^{\text{target}}(\boldsymbol{\theta}^{(t)}) - \mathcal{L}_j^{\text{target}}(\boldsymbol{\theta}^{(t)})| \leq \epsilon', \quad \forall t. \tag{12}$$

Assuming that the loss functions can be approximated by:

$$\mathcal{L}_i^{\text{target}}(\boldsymbol{\theta}) = \frac{1}{2}d\boldsymbol{\theta}^{\top}\boldsymbol{H}_i d\boldsymbol{\theta} + \boldsymbol{g}_i^{\top} d\boldsymbol{\theta} + c_i, \tag{13}$$

where $c_i$ is the loss of example $i$ at the beginning of fine-tuning, and $d\boldsymbol{\theta}$ is the distance between the parameters of the pretrained model and those during fine-tuning. Similarly for $\mathcal{L}_j^{\text{target}}(\boldsymbol{\theta})$. The loss difference between $i$ and $j$ is:

$$\mathcal{L}_i^{\text{target}}(\boldsymbol{\theta}) - \mathcal{L}_j^{\text{target}}(d\boldsymbol{\theta}) = \frac{1}{2}d\boldsymbol{\theta}^{\top}(\boldsymbol{H}_i - \boldsymbol{H}_j)d\boldsymbol{\theta} + (\boldsymbol{g}_i - \boldsymbol{g}_j)^{\top} d\boldsymbol{\theta} + (c_i - c_j). \tag{14}$$

Given that $|\mathcal{L}_i^{\text{target}}(\boldsymbol{\theta}) - \mathcal{L}_j^{\text{target}}(\boldsymbol{\theta})| \leq \epsilon'$, we can write:

$$\left| \frac{1}{2}d\boldsymbol{\theta}^{\top}(\boldsymbol{H}_i - \boldsymbol{H}_j)d\boldsymbol{\theta} + (\boldsymbol{g}_i - \boldsymbol{g}_j)^{\top} d\boldsymbol{\theta} + (c_i - c_j) \right| \leq \epsilon'. \tag{15}$$

Let us choose two different values, $\boldsymbol{\theta}^{(1)}$ and $\boldsymbol{\theta}^{(2)}$, to generate two inequalities. For $d\boldsymbol{\theta}^{(1)}$, we have:

$$\left| \frac{1}{2}(d\boldsymbol{\theta}^{(1)})^{\top}(\boldsymbol{H}_i - \boldsymbol{H}_j)d\boldsymbol{\theta}^{(1)} + (\boldsymbol{g}_i - \boldsymbol{g}_j)^{\top} d\boldsymbol{\theta}^{(1)} + (c_i - c_j) \right| \leq \epsilon', \tag{16}$$

and for $d\boldsymbol{\theta}^{(2)}$, we have:

$$\left| \frac{1}{2}(d\boldsymbol{\theta}^{(2)})^{\top}(\boldsymbol{H}_i - \boldsymbol{H}_j)d\boldsymbol{\theta}^{(2)} + (\boldsymbol{g}_i - \boldsymbol{g}_j)^{\top} d\boldsymbol{\theta}^{(2)} + (c_i - c_j) \right| \leq \epsilon'. \tag{17}$$

Subtracting these two inequalities, we get:

$$\left| \frac{1}{2}\left((d\boldsymbol{\theta}^{(1)})^{\top}(\boldsymbol{H}_i - \boldsymbol{H}_j)\boldsymbol{\theta}^{(1)} - (d\boldsymbol{\theta}^{(2)})^{\top}(\boldsymbol{H}_i - \boldsymbol{H}_j)d\boldsymbol{\theta}^{(2)}\right) + (\boldsymbol{g}_i - \boldsymbol{g}_j)^{\top}(d\boldsymbol{\theta}^{(1)} - d\boldsymbol{\theta}^{(2)}) \right| \leq 2\epsilon'. \tag{18}$$

$$\left| (d\boldsymbol{\theta}^{(1)})^{\top}(\boldsymbol{H}_i - \boldsymbol{H}_j)d\boldsymbol{\theta}^{(1)} - (d\boldsymbol{\theta}^{(2)})^{\top}(\boldsymbol{H}_i - \boldsymbol{H}_j)d\boldsymbol{\theta}^{(2)} \right| \leq \|\boldsymbol{H}_i - \boldsymbol{H}_j\| \left( \|d\boldsymbol{\theta}^{(1)}\|^2 + \|d\boldsymbol{\theta}^{(2)}\|^2 \right)$$
$$\leq (\|\boldsymbol{H}_i\| + \|\boldsymbol{H}_j\|) \left( \|d\boldsymbol{\theta}^{(1)}\|^2 + \|d\boldsymbol{\theta}^{(2)}\|^2 \right)$$
$$\leq 4CD^2 \tag{19}$$

This gives us:

$$\left| (\boldsymbol{g}_i - \boldsymbol{g}_j)^{\top}(d\boldsymbol{\theta}^{(1)} - d\boldsymbol{\theta}^{(2)}) \right| \leq 2\epsilon' + 2CD^2. \tag{20}$$

Assuming $\|d\boldsymbol{\theta}^{(1)} - d\boldsymbol{\theta}^{(2)}\| \geq d$, we get:

$$\|\boldsymbol{g}_i - \boldsymbol{g}_j\| \leq \frac{2\epsilon' + 2CD^2}{d} = \Delta. \tag{21}$$

$\square$

## A.2 Proof of Corollary 4.2

Without loss of generality, assume we select $k$ example from each cluster and we have $k \leq \min_{j \in [K]} |C_j|$. Then the error of the subset in capturing the full gradient will be

$$\xi \leq \sum_j (|C_j| - k)(\bar{\boldsymbol{g}}_j + \Delta), \tag{22}$$

where $\bar{\boldsymbol{g}}_j$ is the norm of the average gradient of the selected examples from $C_j$. In practice, we can weight elements of the subset by $r_{\min}/k$, which has a similar effect to scaling the step size when training on the subset. Let $\boldsymbol{g}_{\max} = \max_j \|\boldsymbol{g}_j\|$ be the maximum gradient norm during training, $r_{\max} = \max_j |C_j|, r_{\min} = \min_j |C_j|$. Then, we get

$$\xi' \leq \sum_j (r_{\min} - k)\Delta + (|C_j| - r_{\min})(\bar{\boldsymbol{g}}_j + \Delta) \tag{23}$$

$$\leq K[r_{\min}\Delta + (r_{\max} - r_{\min})\boldsymbol{g}_{\max}] \tag{24}$$

The first term in RHS of Eq (23) is the error of the subset selected from $C_j$ to capture its full gradient and the second term is due to selecting the same number of examples, $k$, from the larger clusters.

Using the above error and following the proof of Theorem 1 in [35], for a constant step size $\alpha \leq 1/c$ we get:

$$\|\boldsymbol{\theta}^{t+1} - \boldsymbol{\theta}^*\|^2 \leq (1 - \alpha c)^{t+1}\|\boldsymbol{\theta}^t - \boldsymbol{\theta}^*\|^2 + 2\xi' R/c^2 + \alpha B^2 (r_{\min}/k)^2 \boldsymbol{g}_{\max}^2, \tag{25}$$

where $c \leq \|\boldsymbol{H}\|$, and $B = k \cdot K$ is the total size of the subset, $R = \min\{d_0, B\boldsymbol{g}_{\max} + \xi'/c\}$ and $d_0 = \|\boldsymbol{\theta}^0 - \boldsymbol{\theta}^*\|$ is the initial distance to the optimal solution $\boldsymbol{\theta}^*$.

If $k \geq |C_j|$ for any cluster $C_j$, one can simply add $(r_{\min}/k - 1) \cdot \hat{\boldsymbol{g}}_j$ to $\xi'$ for the corresponding clusters, where $\hat{\boldsymbol{g}}_j$ is the norm of the total gradient of cluster $C_j$ and we replace $r_{\min}$ in Eq (23) with the size of smallest cluster that has larger than $k$ examples.

# B Experiment Details

## B.1 Models

**Pythia.** The Pythia models [6] are a suite of large language models (LLMs) developed by EleutherAI licensed under the Apache License 2.0. These models range in size from 70 million to 12 billion parameters and are designed to enable controlled scientific research on transparently trained LLMs across various scales.

**Phi.** The Phi models [28] developed by Microsoft are under the MIT License. Phi-1.5, a transformer-based model with 1.3 billion parameters, and its successor, Phi-2, with 2.7 billion parameters, have been trained on a diverse set of data sources, including synthetic texts and curated websites. The Phi models underscore the potential of small yet powerful language models in understanding and generating human language, empowering a range of NLP tasks. Phi-2, in particular, has raised the bar for reasoning and language understanding among foundation models, matching or even exceeding the performance of models 25 times its size on complex benchmarks.

**LLaMA 2.** The LLaMA 2 models [50], released by Meta AI and licensed under the LLaMA 2 Community License Agreement, are designed for improved natural language understanding and generation. LLaMA 2-7B, the smallest in this series with 7 billion parameters, has demonstrated competitive performance across various NLP benchmarks despite its moderate size.

## B.2 Datasets

**MathInstruct.** The MathInstruct dataset [64] is compiled from 13 diverse math rationale datasets, using both chain-of-thought (CoT) and program-of-thought (PoT) rationales. It ensures comprehensive coverage across various mathematical fields in the 262K training examples, making it a popular resource for fine-tuning large language models (LLMs) for general math problem-solving. MathInstruct is licensed under the MIT license.

Table 3: A synthetic radiology report (MRI of the brain), generated by the GPT-4 model [2] to demonstrate the typical data format and content used in the clinical text summarization task. It is not suitable for clinical or diagnostic use.

| Findings | The brain parenchyma demonstrates normal morphology with no evidence of mass effect or midline shift. No acute infarcts are seen on diffusion-weighted images. There are no signs of intracranial hemorrhage. Mild generalized cerebral atrophy is noted. The ventricles and sulci appear within normal limits for the patient's age. The pituitary gland and sella turcica are unremarkable. There are no abnormal signal intensities within the brain parenchyma. The orbits, paranasal sinuses, and mastoid air cells are clear. |
|---|---|
| Impression | Normal MRI of the brain. Mild cerebral atrophy, likely age-related. No acute intracranial pathology. |

**MIMIC-III.** The MIMIC-III (Medical Information Mart for Intensive Care III) dataset [21] is a comprehensive collection of de-identified health data associated with over 40,000 patients who stayed in critical care units of the Beth Israel Deaconess Medical Center in Boston, Massachusetts. This large dataset includes information such as demographics, vital signs, laboratory tests, medications, and more, making it an invaluable resource for a wide range of research in healthcare, including clinical decision support systems, medical procedure efficacy studies, and patient care optimization strategies.

The MIMIC-III dataset is made freely available to the research community under the Health Insurance Portability and Accountability Act (HIPAA) compliance, ensuring patient confidentiality and data protection. Access to the dataset is granted under a data use agreement (DUA) to individuals affiliated with an institution that approves the use of the data for research purposes. Researchers seeking to utilize the MIMIC-III dataset must complete a required training course on human research protections, which ensures that all researchers are aware of the responsibilities involved in handling sensitive patient data.

### B.3  Implementation Details

**S2L**  The training trajectories for both MathInstruct and MIMIC-III are gathered from training a Pythia-70M model, the smallest model in the Pythia model suite, recorded every $500$ training iterations. We utilize the Faiss library [17] to perform efficient K-means clustering of loss trajectories with Euclidean distance with $K = 100$ and $20$ iterations. The hyperparameter $K$ is tuned in the range of $\{50, 100, 200\}$ based on the average accuracy of the model trained on $30K$ selected data. We found $K = 100$ worked the best for both datasets we studied in this paper. Ablations studies on the length and the best time in the training to record the trajectories can be found in Section 5.4.

**Comparing Reference Models for the Baselines**  For one-shot selection methods (excluding S2L), we use representations from either step 1000 or the end of fine-tuning Pythia-410M on MathInstruct and reported the better result in Figure 4 and Table 1. In Table 4, we include the complete comparison between using early-fine-tuning vs. end-of-fine-tuning model checkpoints as the inference model. For Facility Locations, we further compared using the first hidden states as the feature representation as suggested in [4] to using the last hidden states [56] for the tasks we studied.The ranges for confidence, perplexity, and learnability are chosen according to the best-performing intervals reported in prior research (Section 5.1).

Due to memory and computational constraints, for Facility Locations, we calculate pairwise similarity and perform greedy selection on a per-data-source basis. We found this per-source selection approach also yields benefits for S2L as different data sources within MathInstruct exhibit distinct common patterns in their training trajectories. Therefore, we implement S2L also on a per-source basis for MathInstruct, and recommend applying S2L per source when dealing with datasets composed of multiple data sources.

**Hyperparameters**  Following the setup used in [64], we adopt a training regimen with a learning rate of 2e-5, a batch size of 128, a maximum length of 512, and a cosine scheduler with a 3% warm-up period.

**Experiments Compute Resources**    We fine-tune all the models with the Huggingface transformers library [55] with Fully Sharded Data Parallel (FSDP) [66] on 4 48G NVIDIA RTX A6000.

Table 4: Complete results used for selecting the best reference model for each one-shot data selection baseline. The choice of early-fine-tuning (step 1000) and end-of-fine-tuning checkpoint follows [34]. The best results selected for Figure 4 are highlighted in cyan.

| SELECTION | REF MODEL | DATA SIZE | IN-DOMAIN | | | | OUT-DOMAIN | | | |
|---|---|---|---|---|---|---|---|---|---|---|
| | | | GSM8K | MATH | NUMGLUE | AVG | SVAMP | MATHEMATICS | SIMULEQ | AVG |
| LEAST CONFIDENCE | EARLY | 30K | 2.3 | 1.7 | 15.5 | 6.5 | 13.6 | 1.2 | 0.5 | 5.8 |
| | | 50K | 1.7 | 2.6 | 20.5 | 8.3 | 16.0 | 4.0 | 1.8 | 7.8 |
| | | 100K | 3.9 | 2.7 | 22.5 | 9.7 | 19.2 | 8.0 | 3.3 | 9.9 |
| | END | 30K | 2.7 | 1.3 | 18.0 | 7.0 | 13.7 | 3.3 | 1.4 | 6.7 |
| | | 50K | 2.1 | 1.7 | 21.0 | 8.3 | 14.5 | 3.5 | 1.0 | 7.3 |
| | | 100K | 2.5 | 3.3 | 23.5 | 9.8 | 20.8 | 6.3 | 3.7 | 10.0 |
| MIDDLE PERPLEXITY | EARLY | 30K | 3.3 | 3.8 | 17.5 | 8.2 | 11.8 | 1.2 | 1.2 | 6.5 |
| | | 50K | 2.9 | 4.1 | 19.6 | 8.9 | 15.6 | 7.6 | 2.9 | 8.8 |
| | | 100K | 4.8 | 7.1 | 20.4 | 10.8 | 19.6 | 16.1 | 3.9 | 12.0 |
| | END | 30K | 5.3 | 3.7 | 16.2 | 8.4 | 14.2 | 8.7 | 1.2 | 8.2 |
| | | 50K | 3.2 | 5.9 | 20.5 | 9.9 | 18.1 | 11.3 | **5.1** | 10.7 |
| | | 100K | 5.4 | 7.2 | 20.9 | 11.2 | **23.8** | 15.3 | 3.3 | 12.6 |
| HIGH LEARNABILITY | EARLY | 30K | 6.1 | 1.6 | 19.1 | 8.9 | 10.7 | 9.9 | 1.4 | 8.1 |
| | | 50K | 6.1 | 2.1 | 18.6 | 8.9 | 14.5 | 14.0 | 2.1 | 8.9 |
| | | 100K | **7.4** | 9.2 | **29.8** | **15.5** | 20.7 | 19.4 | **10.3** | **16.1** |
| | END | 30K | 3.0 | 1.4 | 14.7 | 6.4 | 2.1 | 6.8 | 1.8 | 5.0 |
| | | 50K | 1.3 | 2.1 | 16.0 | 6.5 | 4.7 | 6.9 | 3.1 | 5.7 |
| | | 100K | 4.3 | 7.2 | 23.0 | 11.5 | 16.7 | 16.1 | 4.3 | 11.9 |
| FACILITY LOCATION | EARLY (FIRST) | 50K | 3.9 | 7.6 | 12.4 | 8.0 | 11.1 | 14.6 | 1.9 | 8.6 |
| | EARLY (LAST) | 50K | 5.7 | 9.1 | 12.4 | 9.1 | 15.4 | 18.6 | 1.6 | 10.5 |
| | END (FIRST) | 50K | 3.8 | 7.7 | 14.8 | 8.7 | 19.2 | 11.4 | 2.3 | 9.9 |
| | END (LAST) | 50K | 5.2 | **9.7** | 11.8 | 8.9 | 12.4 | 18.2 | 1.0 | 9.7 |

## B.4   Evaluation

### B.4.1   MathInstruct

**Datasets.**    We utilize 6 diverse datasets with open-formed questions for evaluating the mathematical reasoning capabilities of models trained with both the full MathInstruct dataset and selected subsets. These datasets, detailed in Table 5, span a range of mathematical disciplines from early algebra to calculus and linear algebra, covering various types of questions such as multi-step reasoning, arithmetic word problems, and problems from mathematics competitions. This variety ensures a comprehensive assessment across both in-domain and out-domain tasks.

**Pipeline.**    We utilize the pipeline provided by [64][2], designed to first determine whether the model can be prompted to generate a code snippet. This code snippet, if successfully generated, should be executable and produce the correct answer when run. This code-based evaluation is also used for Phi models [28]. In cases where the model does not directly produce a viable code solution, we employ a "think step-by-step" prompting strategy [54]. This method prompts the model to break down its reasoning process, a technique that has been widely proven effective in fully exploiting the model's problem-solving capacity.

### B.4.2   MIMIC-III

Following [14, 15], we include the six most common modality/anatomy pairs: CT head, CT abdomen, CT chest, MRI head, CT spine, and CT neck, and five less common pairs in the text data: MRI spine, CT sinus, MRI abdomen, MRI pelvis, and MRI neck in the evaluation. There are in total 13.7K test examples after data preprocessing and train-test splitting.

---

[2]https://github.com/TIGER-AI-Lab/MAmmoTH?tab=readme-ov-file#large-scale-evaluation

Table 5: Types of questions in the evaluation datasets for the mathematical reasoning task.

| DATASET | SIZE | LEVEL | TASKS |
|---|---|---|---|
| GSM8K | 1319 | Early Algebra | Multi-step reasoning using basic arithmetic operations |
| MATH | 5000 | Early Algebra, Intermediate Algebra, Algebra, Probability, NumTheory, Calculus, Geometry | Problems from mathematics competitions, including the AMC 10, AMC 12, AIME |
| NumGLUE | 1042 | Early Algebra | Commonsense, Domain-specific, Arithmetic Reasoning, Quantitative Comparison, Fill-in-the-blanks Format, Reading Comprehension, Numerical Reasoning, Quantitative NLI, Arithmetic Word Problems |
| SVAMP | 1000 | Early Algebra | Arithmetic Word Problems |
| Mathematics | 1000 | Early Algebra, Intermediate Algebra, NumTheory, Calculus | Arithmetic Reasoning |
| SimulEq | 514 | Linear Algebra | Single and multiple equation word problems |

# C  Examples in Different Clusters

We compare data points in the same and different clusters based on training trajectories, in Figure 15, Figure 16 and Figure 17. We can observe that examples with similar training trajectories have the same question format. Therefore, balanced sampling from all clusters can ensure different types of examples can be covered in the selected subset of training data.

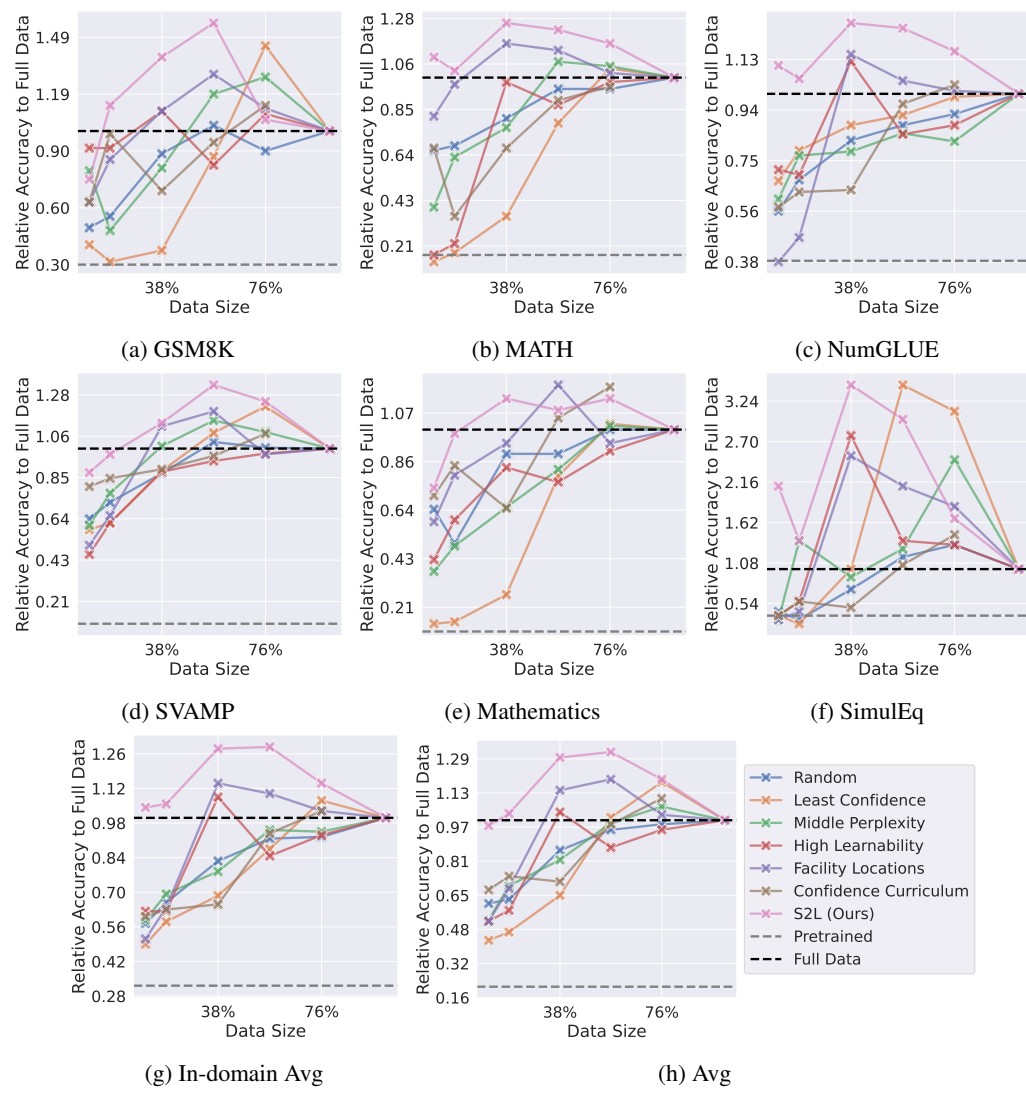

Figure 14: Break-down accuracies (↑) on in-domain and out-of-domain datasets using Pythia-410M. Data size refers to the total number of unique training examples used. All experiments in this table share the same total training steps and learning rate schedule (see Section 5.2).

# D    Topic Distribution of Data Selected by S2L

Beyond qualitative examples from different clusters, we study how S2L changes the data distribution to outperform using the full fine-tuning dataset as well as using random subsets of the same size that have the same distribution as the original dataset. In Figure 18, we can observe that S2L not only guarantees a thorough and balanced coverage across the spectrum of topics but also ensures sufficient representation of foundational topics, such as pre-algebra, which lays the groundwork for tackling more complex subjects.

# E    Broader Impacts

This paper introduces a data selection method for large language models (LLMs), aiming to enhance the data efficiency in the supervised fine-tuning (SFT) of these models.

Camden went swimming 16 times in March and Susannah went 24 times. If the number of times they went throughout the month was divided equally among 4 weeks, how many more times a week did Susannah swim than Camden?
Camden went swimming 16/4 = 4 times a week
Susannah went swimming 24/4 = 6 times a week
Susannah went 6 - 4 = 2 more times a week than Camden
The answer is 2

· · · · · · · · · · · · · · · · · · · · · · · · · · · · · · · · · · · · · · · · · · · · · · ·

What is the perimeter of a Koch snowflake curve after five iterations, if the initial side length of the equilateral triangle used to construct the curve is 8 cm?
The Koch snowflake is a fractal curve constructed by iteratively replacing each side of an equilateral triangle with a smaller equilateral triangle. After each iteration, the number of sides increases by a factor of 4, and the length of each side is reduced by a factor of 3.
After five iterations, the number of sides in the Koch snowflake is given by:
number of sides = 3 * (4^5) = 3 * 1024 = 3072
The length of each side after five iterations is given by:
side length = initial side length / (3^5) = 8 / 243
The perimeter of the Koch snowflake after five iterations is the product of the number of sides and the length of each side:
perimeter = number of sides * side length = 3072 * (8 / 243) ≈ 101.3333 cm

· · · · · · · · · · · · · · · · · · · · · · · · · · · · · · · · · · · · · · · · · · · · · · ·

Lilly and Fiona are cleaning a room. Between them, it takes 8 hours to clean the room. A quarter of the time spent cleaning was by Lilly and Fiona was responsible for the rest of the cleaning. How long, in minutes, was Fiona cleaning?
8 hours is 8*60=480 minutes.
Lilly cleaned for 480/4=120 minutes.
This means that Fiona was cleaning for 480-120=360 minutes.
The answer is 360

Figure 15: Examples in the cluster shown in Figure 7a: open-formed algebra. Questions are in black and answers are in cyan.

Passage: The Raiders began their 2011 campaign at Sports Authority Field at Mile High, for a Week 1 AFC West duel with the Denver Broncos in the second game of Monday Night Football's doubleheader. Oakland trailed early in the first quarter as Broncos kicker Matt Prater got a 28-yard field goal. The Raiders answered in the second quarter as quarterback Jason Campbell found fullback Marcel Reece on a 3-yard touchdown pass, followed by a 37-yard, a 21-yard, and an NFL record tying 63-yard field goal from kicker Sebastian Janikowski. Janikowski's leg helped put the Raiders up 16-3 at halftime. Denver answered in the third quarter as wide receiver Eric Decker returned a punt 90 yards for a touchdown, followed by Prater getting a 30-yard field goal. Oakland struck back in the fourth quarter with Campbell's 1-yard touchdown. The Broncos tried to rally with quarterback Kyle Orton completing a 9-yard touchdown pass to running back Lance Ball, yet the Raiders' offense was able to run out the clock. With the win, not only did Oakland begin their season at 1-0, but they also snapped their 8-straight opening day losing streak. Question: How many yards was the second longest field goal?
Let's write a program.
second = 37
print(second)

· · · · · · · · · · · · · · · · · · · · · · · · · · · · · · · · · · · · · · · · · · · · · · ·

Passage: The U.S. Institute of Medicine (IOM) updated Estimated Average Requirements (EARs) and Recommended Dietary Allowances (RDAs) for iron in 2001. The current EAR for iron for women ages 14–18 is 7.9 mg/day, 8.1 for ages 19–50 and 5.0 thereafter (post menopause). For men the EAR is 6.0 mg/day for ages 19 and up. The RDA is 15.0 mg/day for women ages 15–18, 18.0 for 19–50 and 8.0 thereafter. For men, 8.0 mg/day for ages 19 and up. RDAs are higher than EARs so as to identify amounts that will cover people with higher than average requirements. RDA for pregnancy is 27 mg/day and, for lactation, 9 mg/day. For children ages 1–3 years 7 mg/day, 10 for ages 4–8 and 8 for ages 9–13. As for safety, the IOM also sets Tolerable upper intake levels (ULs) for vitamins and minerals when evidence is sufficient. In the case of iron the UL is set at 45 mg/day. Collectively the EARs, RDAs and ULs are referred to as Dietary Reference Intakes. Question: How many years does an RDA of 8 last for children?
Let's write a Python program to solve it.
child = 4
print(child)

Figure 16: Examples in the cluster shown in Figure 7b: reading comprehension + coding. Questions are in black and answers are in cyan; instructions are highlighted in orange.

**Positive Impacts:** Our method, by reducing the data requirements for training LLMs, can make fine-tuning LLMs more effective and accessible. This could lead to broader participation in AI research and application development across various fields, including healthcare and education.

**Negative Impacts:** Our method does not inherently involve or encourage applications with direct negative societal impacts. The focus is on a generic improvement in the field of machine learning, particularly in the training of LLMs.

**If there are n rays and all of them are reversed after 2006 reversals, then a possible value of n is**
**Answer Choices: (A) 2237 (B) 3777 (C) 2002 (D) 3772 (E) 8822**
Let's reason about the multi-choice question.
has to be same number less than (or equal to) k and of the same type as k.
k = 2006 then n = 2002.
The answer is C

. . . . . . . . . . . . . . . . . . . . . . . . . . . . . . . . . . . . . .

**Stephanie, Regine, and Brian ran a 20 mile race. Stephanie and Regine's combined times exceeded Brian's time by**
**exactly 2 hours. If nobody ran faster than 8 miles per hour, who could have won the race?**
**I. Regine**
**II. Stephanie**
**III. Brian**
**Answer Choices: (A) I only (B) II only (C) III only (D) I or II only (E) II, or III**
Let's solve the multi-choice question step by step.
Given that S+R=B+2, where S, R, and B are times in which Stephanie, Regine, and Brian completed the race.
Min time one could complete the race is 20/8=2.5 hours. Let's see if Brian could have won the race: if he ran at the
fastest rate, he would complete the race in 2.5 hours, so combined time needed for Stephanie and Regine would be
S+R=B+2=4.5 hours, which is not possible as sum of two must be more than or equal the twice the least time:
2*2.5=5. So Brian could not have won the race.
There is no reason to distinguish Stephanie and Regine so if one could have won the race, another also could. So
both could have won the race.
The answer is E.

. . . . . . . . . . . . . . . . . . . . . . . . . . . . . . . . . . . . . . .

**If m is the average (arithmetic mean) of the first 10 positive multiples of 4 and if M is the median of the first 10**
**positive multiples of 4, what is the value of M − m ?**
**Answer Choices: (A) 0 (B) -5 (C) 5 (D) 25 (E) 27.5**
Let's think about the multi-choice question step by step.
The first 10 positive multiples of 4 is an evenly spaced set. One of the most important properties of evenly spaced
set (aka arithmetic progression) is:in any evenly spaced set the arithmetic mean (average) is equal to the median.
Hence M=m --> M-m=0.
The answer is A.

Figure 17: Examples in the cluster shown in Figure 7c: multiple-choice + multi-step reasoning. Questions are in black and answers are in cyan; instructions are highlighted in orange.

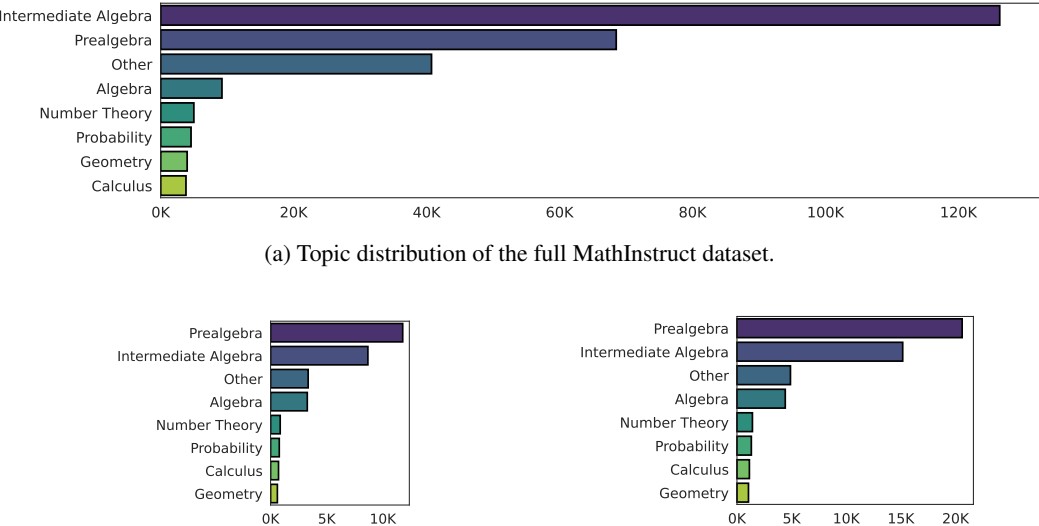

(a) Topic distribution of the full MathInstruct dataset.

(b) Topic distribution of 30K data selected by S2L.    (c) Topic distribution of 50K data selected by S2L.

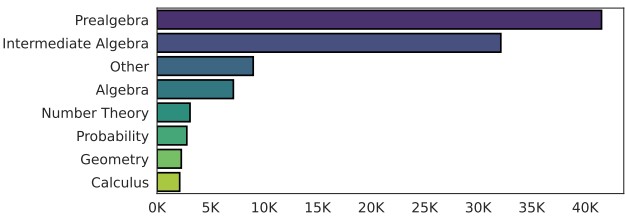

(d) Topic distribution of 100K data selected by S2L.

Figure 18: Compared to the original topic distribution, S2L prioritized easier topics (e.g., pre-algebra over intermediate algebra, algebra over other more advanced topics) while always ensuring complete and more balanced coverage of all topics.

