# OpenReview forum: "SmallToLarge (S2L): Scalable Data Selection for Fine-tuning Large Language Models by Summarizing Training Trajectories of Small Models"
_NeurIPS.cc/2024/Conference — NeurIPS 2024 poster_

### Official Review · Reviewer_dErU · 2024-06-19

**Soundness:** 3
**Presentation:** 3
**Contribution:** 2
**Rating:** 4
**Confidence:** 3

**Summary:**

This paper proposes a data selection approach to reduce the sample size required to conduct supervised fine-tuning (SFT) of LLMs for specific domains. The method achieves it by approximating the training gradients on the full data using only a subset of data.

The experiments were done for SFT tasks, including (1) math problem-solving and (2) clinical text summarization. It was found that with 11% of the original dataset, the yielded model can be comparable to the one trained on the full dataset.

**Strengths:**

The paper provides a clear rationale for selecting data for Sparse Fine-Tuning (SFT) by estimating the full gradients with a subset of samples. It conducts a thorough analysis of various variants of the proposed method and includes experiments across diverse scenarios of its application.

**Weaknesses:**

1. Data selection for machine learning has been explored in the literature. Some related works which this paper omits but may need to compare and discuss include:

- influence function based approaches [1,2]

- reinforcement learning based approaches [3]

- data Shapley [4]

2. The generalizability of the proposed theory remains elusive. A small model is used to approximate the gradients. There are two gaps

- The small model can have a distinct loss landscape from the large model. Approximating full gradients of a small model not necessarily extrapolate to a large and different models.

- There is no theoretical guidance on how this small model should be like, which architecture, which parameter size, to be competent of approximating the gradients of the large model.

Considering the black-box nature of deep learning, especially for LLMs, it seems very hard to build such theoretic foundation for the proposed method.


[1] Wang, Z., Zhu, H., Dong, Z., He, X., & Huang, S. L. (2020, April). Less is better: Unweighted data subsampling via influence function. In Proceedings of the AAAI Conference on Artificial Intelligence (Vol. 34, No. 04, pp. 6340-6347).

[2] Kong, S., Shen, Y., & Huang, L. (2021, October). Resolving training biases via influence-based data relabeling. In International Conference on Learning Representations.

[3] Yoon, J., Arik, S., & Pfister, T. (2020, November). Data valuation using reinforcement learning. In International Conference on Machine Learning (pp. 10842-10851). PMLR.

[4] Ghorbani, A., & Zou, J. (2019, May). Data shapley: Equitable valuation of data for machine learning. In International conference on machine learning (pp. 2242-2251). PMLR.

**Questions:**

From Figure 4, it is surprising that the proposed method and some baselines can outperform the full data performance with around 38% of the samples. How much benefit can the method bring if we continue to scale the data?

**Limitations:**

do not apply

---

> ### Author Rebuttal · Authors · 2024-08-07
>
> Thank you for your feedback. We appreciate your positive comments on the rationale and thorough analysis of our method. We address your concerns and provide clarifications below:
>
> #### **1. Omitted Related Works:**
> Our paper includes related works in the data selection domain, but we will expand this section to incorporate a discussion of the specific references you mentioned in our future revisions.
>
> Influence function-based approaches, such as those by Wang et al. (2020) and Kong et al. (2021), are not directly applicable to the problem we address with S2L. These methods rely on the availability of a validation set that is a very good representative of the test set, as they find training examples most similar to the validation examples. This is a different problem from ours, which does not assume the availability of a good validation set.
>
> Similarly, reinforcement learning-based approaches, such as Yoon et al. (2020), also address a different problem from ours. It aims to quantify the value of data by learning data values that improve model performance on a small validation set, focusing on tasks like domain adaptation. This approach also requires a clean validation set that is a good representative of the test set.
>
> In contrast, S2L focuses on dropping redundancies in the data, ensuring data diversity without the need for a clean and representative validation set. Thus, **influence-based and reinforcement learning-based methods [1,2,3] are not applicable to our problem** and are not considered as baselines.
>
> Calculating Data Shapley values requires evaluating all possible subsets of the dataset, **(for a dataset with $ |D| $ instances, the time complexity of directly calculating Shapley values is $ O(|D| \cdot 2^{|D|}) $,) making it computationally infeasible for large datasets typically used in training LLMs.** Approximations to Data Shapley values reduce the computational burden but still remain impractical for large-scale applications​. Typically, estimating Shapley values accurately might involve several hundred to thousands of permutations, where each permutation involves training a model on different subsets of the data. In contrast, S2L’s training complexity is only a single additional training round of a smaller proxy model and its inference complexity is linear with respect to the proxy dataset size.
>
> #### **2. Generalizability of the Proposed Theory:**
> In our paper, specifically in the "Small-to-Large Data Selection" paragraph in Section 4, we provided references and experiments supporting the use of small proxy models. In Figure 3 of our paper, we show that examples in the same loss trajectory cluster of a small model also have similar loss trajectories on a large model. This demonstrates that the clustering of loss trajectories from a small model can effectively capture the data characteristics relevant for training larger models, thereby validating our approach empirically.
>
> The reviewer is correct that in our proof, we assumed finding the loss trajectories using the target model used for fine-tuning. Assuming that the distance between loss-trajectories of examples on the proxy and target models is bounded by some constant, this can be incorporated into our theory to bound the distance between the gradient of the subset and the full data at every step of (incremental) gradient descent, and can be used to bound the size of the neighborhood around the optimal solution (found by training the target model on full data) that the target model converges to when trained on the subset. For the fine-tuning setting where we assume a bounded curvature for the proxy and target models, the above assumption is reasonable. We thank the reviewer and will incorporate this discussion.
>
> Additionally, Xia et al. (2022) analyzed the training trajectories of differently sized OPT models, ranging from 125M to 175B parameters. They showed that models of different sizes within the same architecture family pre-trained with the same data exhibit similar learning patterns and behaviors at equivalent levels of training perplexity, supporting the idea that smaller models can provide valuable insights for larger models. However, theoretically bounding the distance between loss trajectories of such models is not trivial and requires future investigation.
>
>
> ### **Questions:**
> **New results:** Figure 3 of the one-page PDF attached in the global rebuttal **([link](https://openreview.net/attachment?id=lyJkoUGNPM&name=pdf))** shows that S2L first increases in relative accuracy compared to the full data approach and then its advantage over training on full data decreases as data size continues to scale up. Ultimately, as data size approaches 100%, the accuracy will converge to that of training on the full dataset.
>
> Note that we kept the total training iterations/steps consistent for all results in Figure 3, including training on the full dataset. As discussed in the introduction section of our paper, training more on smaller, higher-quality data can be more effective than training on larger, redundant datasets. Even if we train on the full data for more epochs, our experiments show that training on S2L selected data yields higher accuracy (Figure 4 in the one-page PDF attached in the global rebuttal ([link](https://openreview.net/attachment?id=lyJkoUGNPM&name=pdf))) than training on full data with the same number of epochs, even though in this case full data takes more training budget/iterations compared to training with S2L selected subsets. By clustering based on loss trajectories, S2L ensures that the selected data is of high quality and representative, allowing the model to focus on the most informative examples. Results in Figure 3 indicate that **S2L can bring significant benefits in terms of data efficiency, making better use of the available data.**
>
> We hope these clarifications address your concerns. Thank you once again for your valuable feedback.

---

> ### Author Response · Authors · 2024-08-12
>
> Dear Reviewer dErU,
>
> We hope our recent rebuttal has been helpful in addressing your concerns. If you have any remaining questions, please don't hesitate to let us know—we’d be more than happy to discuss further. We truly appreciate the time and effort you’ve put into reviewing our work, so we want to ensure we’ve adequately responded to your feedback.
>
> Best regards,
>
> SmallToLarge (S2L) Authors

---

### Official Review · Reviewer_Ho1K · 2024-07-12

**Soundness:** 3
**Presentation:** 3
**Contribution:** 3
**Rating:** 7
**Confidence:** 5

**Summary:**

This paper introduces a data selection method called Small to Large (S2L), which uses the training trajectory of a small model to build clusters and select data based on these clusters. The method is shown to be effective in two domains, math and medicine, with superior performance than other data selection methods and, oftentimes, better than training on the full dataset. A theoretical guarantee is also provided for the convergence of training based on this data selection method. Extensive analysis demonstrates S2L's effectiveness under various compute budgets and across different model series.

**Strengths:**

- The contribution of this work is indeed unique, as most data selection methods in the LLM era are focused on initial or continual pretraining and IFT, with relatively sparse attention given to SFT data in specific domains. This is important for many real-life LLM applications.
- The idea is intuitive, as the training trajectory is informative of data characteristics, and effective. Additional theoretical support is provided for the similar effects during training by in-cluster data and the convergence analysis of data selected by S2L. Extensive experiments and analysis demonstrate the overall effectiveness of this method across various compute budgets (Figure 4) and for various model families like Pythia, Phi, and Llama.
- Extensive analysis is also being conducted to better understand the S2L method, specifically the synergy between the cluster and fine-tuned model's embedding in terms of data similarity, as well as the robustness across the length and time frame of the training trajectory.

**Weaknesses:**

- As shown in Figure 4, it is clear that for many methods, the SFT has not yet converged. It would be interesting to see how each method affects the convergence of the SFT process and what the relative accuracy looks like, i.e., whether better performance can be further achieved.
- The work studies the Pythia model as the small reference model and multiple models from different model families for task evaluation. It is intriguing to know whether other small models can serve as good reference models, and how the alignment between the pretraining data of the small and large models will affect the effectiveness of this data selection strategy.
- In the LLM era, the size of SFT data scales up quickly. It will be interesting to see how this method scales with the SFT data size, for instance, at M/B token scale.
- It is still tricky to determine whether the effect of such clusters will be influenced by different learning rate schedulers and optimizers.

**Questions:**

Please refer to weaknesses.

**Limitations:**

The work addresses the limitations of the limited domain tested and the limited size (7B) of the model used for evaluation.

---

> ### Author Rebuttal · Authors · 2024-08-07
>
> Thank you for your detailed review and constructive feedback on our paper. We appreciate your positive comments on the uniqueness, intuitiveness, theoretical support, and extensive experiments of our study. We address your concerns and provide clarifications below:
>
> #### **1. Convergence of SFT:**
> The x-axis of Figure 4 is the size of the subset that we trained on till convergence. We conducted additional experiments extending our analysis to larger data sizes. Due to time and resource constraints, we focused on two data sizes larger than 38% (57% and 76%) for this extended analysis and 3 best-performing baselines in our original submission. The results are presented in Figure 3 in the one-page PDF attached in the global rebuttal ([link](https://openreview.net/attachment?id=lyJkoUGNPM&name=pdf)), which extends Figure 4 from our main submission.
> We observe that:
> - Random monotonically approaches full data performance as the data size increases, showing expected behavior.
> - High Learnability does not show improvement when scaling beyond 38% of the data.
> - Both Facility Locations and our S2L method continue to improve when increasing from 38% to 57% of the data.
> - As we approach 100%, the performance of all methods drops and converges to the dashed line (full data).
>
> Meanwhile, S2L consistently outperforms other approaches across all data sizes, both for in-domain and overall average accuracy.
> In our revision, we plan to include the remaining methods from Figure 4 that are not present in this new Figure 3.
>
> #### **2. Using Other Small Reference Models:**
> **New results:** We used GPT-2 (2019), which has 124M parameters, and compared it to Pythia-160M (2024) as the reference model to select data to train Pythia-410M. GPT-2 was pretrained on the WebText dataset, which consists of 8 million web pages scraped from outbound links on Reddit, and cleaned to remove Wikipedia pages and duplicates. The Pythia models, on the other hand, were trained on the Pile dataset, a curated collection of English language datasets that is popular for training large autoregressive transformers. The Pile dataset contains approximately 300 billion tokens and includes diverse sources such as books, Wikipedia, and various web texts. Results in Figure 5 in the one-page PDF attached in the global rebuttal ([link](https://openreview.net/attachment?id=lyJkoUGNPM&name=pdf)) show that both proxy models perform comparably in guiding the data selection for training Pythia-410M, demonstrating that different small models can be effectively used as reference models.
>
> #### **3. Scaling with SFT Data Size:**
> **New results:** The MathInstruct dataset, which we used in our experiments, contains approximately 60M tokens. Our method has shown strong performance and scalability with this dataset size.
>
> To further improve scalability, we conducted experiments by training the proxy on a smaller sample of the data (100K examples, ~30M tokens) for the same number of epochs (3 epochs) and saving the loss for all examples. Figure 1 of the one-page PDF attached in the global rebuttal ([link](https://openreview.net/attachment?id=lyJkoUGNPM&name=pdf)) presents the results of these experiments,  demonstrating its efficiency and scalability for large datasets.
>
> #### **4. Effect of Different Learning Rate Schedulers and Optimizers:**
> We tuned hyperparameters for fine-tuning the target model, specifically learning rate $\in$ {2e-5, 5e-6} and number of training epochs $\in$ {2, 3, 4}. Standard values were used for warmup ratio (0.03), weight decay (0), and learning rate scheduler type (cosine) as per [64]. The same hyperparameters were applied for all data selection methods, including S2L and the baselines, ensuring a fair comparison. For S2L, these hyperparameters were consistently used for both training the proxy model and fine-tuning the target model.
>
> We hope these clarifications address your concerns. Thank you once again for your valuable feedback.

---

> > ### Comment · Reviewer_Ho1K · 2024-08-10
> >
> > Thanks to the authors for the detailed reply. All of my concerns have been addressed.

---

> > > ### Author Response · Authors · 2024-08-10
> > >
> > > Thank you very much for taking the time to carefully review our rebuttal. We’re glad to hear that our responses addressed your concerns and appreciate the increase in your rating and confidence.
> > >
> > > Your feedback has really helped us improve our work. Thanks again for your valuable input.

---

### Official Review · Reviewer_5vbK · 2024-07-13

**Soundness:** 3
**Presentation:** 4
**Contribution:** 2
**Rating:** 5
**Confidence:** 4

**Summary:**

This study presents an innovative technique aimed at improving data efficiency in the supervised fine-tuning of large language models for niche domains. The method leverages the training trajectories of smaller models to inform the data selection for larger models, thereby maximizing the utility of the available data. Experimental outcomes on tasks such as mathematical problem-solving and clinical text summarization demonstrate that this technique can significantly reduce the training dataset size while delivering better performance than other standard approaches.

**Strengths:**

- The paper addresses a critical objective: enhancing the efficiency of large models through data selection, utilizing a smaller model to streamline the selection process.
- Extensive experiments are conducted to demonstrate the effectiveness of this approach.
- The paper is well-structured, with the method and results clearly presented and supported by theoretical analysis.

**Weaknesses:**

- Overclaim: The paper showcases experiments on only two specialized domain datasets, yet it is presented as if the method is universally applicable. While the introduction offers some context, the title remains overly broad, and the methodology lacks domain-specific adaptations. The authors should refine the framing of their contribution to more accurately reflect its actual scope.

- Marginal Novelty: Data selection using proxy models is not a new concept, as highlighted by reference [12]. Additionally, the use of training trajectories has been explored in prior studies, such as [1].

- Limited Evaluation: It would be beneficial to conduct experiments with larger language models beyond the 7B scale to validate the proposed method's effectiveness. The results only suggest that the method may be useful for smaller language models. Also, it would be better to calculate the win-rate for the summarization dataset.

[1] LESS: Selecting Influential Data for Targeted Instruction Tuning

**Questions:**

1. I am still not sure why using a small fraction of data is better than the performance of using full data on NUMGLUE and MATHEMATICS. Can authors provide more insights on this?
2. How to determine the number of K for clustering?

**Limitations:**

Yes

---

> ### Author Rebuttal · Authors · 2024-08-07
>
> Thank you for your feedback. We appreciate your positive comments on the objective, experiments, structure, and theoretical analysis. We address your concerns and provide clarifications below:
>
> #### **1. Overclaim:**
> S2L is generally applicable to various domains without requiring specific adaptations, ensuring its broad usability. As long as the fine-tuning data is in the form of pairs of questions and answers, our method is expected to work without any further modification. Converting domain-specific data (e.g., EHR records) into question/answer pairs has been studied in the literature and is beyond the scope of our work. Our study focuses solely on the "fine-tuning step" after the input data is formatted into question/answer pairs. We will clarify this point in our revised manuscript.
>
> Our title specifies that the method is aimed at improving data efficiency in supervised fine-tuning for specialized domains, which accurately reflects the scope of our work. Our experiments were conducted on more than one specialized domain (mathematical problem-solving and clinical text summarization) to demonstrate the potential versatility of S2L. We agree that further validation in additional domains is necessary and will ensure that the revised manuscript emphasizes this point.
>
> #### **2. Marginal Novelty:**
> While proxy models are used by prior work such as [12], those approaches _do not translate well to LLMs_. They employ metrics like forgetting scores, uncertainty, or representations of a proxy model for data selection. Forgetting scores track the number of times a model correctly classifies examples and then incorrectly classifies them in the subsequent training iteration. While this approach is well-defined for image classification, it does not translate well to autoregressive LLMs because they do not do sample-level classification. Uncertainty metrics, which measure the model’s confidence in its predictions, as well as using representations were included as our baselines and we showed that S2L is more effective than these methods. Moreover, these methods rely on heuristics that do not provide theoretical insights.
>
> Due to the above-mentioned reasons, **using a proxy model to select data for LLMs requires a different approach that works and scales better for LLMs than existing ones, which is the main contribution of our work.** Unlike prior heuristics, we have provided a **theoretically justified framework** by leveraging the theoretical observation that examples with similar loss trajectories have similar gradients during training and empirically demonstrating that loss trajectories can effectively guide data selection.
>
> Regarding LESS, it _addresses a different problem and is not applicable to our setting._ LESS selects training examples whose gradients are most similar to the gradients of a validation set, focusing on targeted instruction tuning. In contrast, **S2L does not assume access to a validation set and works without the need for clean validation data.** This makes the problems S2L can address potentially more difficult. Moreover, LESS requires projecting gradients into a low-dimensional space, which is computationally expensive and slow. S2L, on the other hand, leverages loss trajectories (which are **easier and faster to compute**) collected from training a smaller model and uses clustering to identify representative subsets, focusing on balanced representation across clusters to ensure data diversity and efficiency.
>
> #### **3. Limited Evaluation:**
> As an academic research lab, our computational resources are limited, and models at the scale of 7B parameters represent the largest scale we can currently experiment with. Our 7B experiments with a batch size of 64 and a maximum sequence length of 512 require 48G memory per GPU to host on 8 48G NVIDIA A6000 GPUs. Hosting models up to the 70B scale is much beyond our computational capacity. According to `accelerate`'s estimate-memory tool, training `meta-llama/Llama-2-70b-hf` with a batch size of 1 using Adam optimizer and fp32 requires 1TB GPU memory, while fp/bp16 requires 512G GPU memory, far exceeding our resources. Nevertheless, we believe that our results on models up to 7B are significant and provide a strong indication of the method’s effectiveness.
>
> **New results:** In response to your suggestion about calculating win-rates, we used GPT-3.5-turbo as an automated judge to evaluate the clinical summaries. We presented the judge LLM with the original findings and impression from the radiology reports and two anonymized summaries (one generated by our model trained with S2L selected data, and another generated by a model trained with full data).
> Our prompt instructed GPT-3.5-turbo to act as an expert radiologist, evaluating the summaries based on accuracy, completeness, relevance, and coherence, while using the original impression as a reference. The win-rate was calculated as the percentage of times the model was preferred. Our S2L model achieved a 54.8% win-rate against the full-data model.
>
> #### **Questions:**
>
> 1. **Performance with a Small Fraction of Data:**
> As discussed in the introduction section of our paper, training more on smaller, higher-quality data can work better than training on larger, lower-quality datasets [48,67]. Our method is theoretically guaranteed to remove redundancy (i.e. examples with highly similar effect on training) to allow the model to learn more effectively from a diverse and representative subset of training data during fine-tuning. By ensuring that the selected data is non-redundant and representative, S2L allows the model to focus on the most informative examples, thereby enhancing the overall training efficiency and performance.
>
> 2. **Determining the Number of Clusters (K):**
> Please see Figure 2 in the global rebuttal ([link](https://openreview.net/forum?id=K9IGlMQpif&noteId=lyJkoUGNPM)).
>
> We hope these clarifications address your concerns. Thank you once again for your valuable feedback.

---

> ### Author Response · Authors · 2024-08-12
>
> Dear Reviewer 5vbK,
>
> We hope our recent rebuttal has been helpful in addressing your concerns. If you have any remaining questions, please don't hesitate to let us know—we’d be more than happy to discuss further. We truly appreciate the time and effort you’ve put into reviewing our work, so we want to ensure we’ve adequately responded to your feedback.
>
> Best regards,
>
> SmallToLarge (S2L) Authors

---

### Official Review · Reviewer_mU1m · 2024-07-13

**Soundness:** 2
**Presentation:** 3
**Contribution:** 3
**Rating:** 5
**Confidence:** 4

**Summary:**

This paper addresses the challenge of data selection in supervised fine-tuning (SFT) of pretrained language models by introducing a novel method called SmallToLarge (S2L). The S2L method involves collecting loss trajectories from a smaller model, clustering these trajectories, and resampling the SFT data to ensure balanced representation across clusters within the given data budget. The authors tested S2L on tasks such as mathematical reasoning and clinical data summarization, demonstrating superior results compared to several existing offline and online data selection methods.

**Strengths:**

1. The motivation of this study is clear, and the proposed S2L method enhances data efficiency in supervised fine-tuning (SFT) for specialized domains.
2. The writing is overall clear and the paper is easy to read.
3. Extensive experiments are conducted to validate the efficacy of the proposed approach.
4. The authors provide detailed analysis about why this method works.

**Weaknesses:**

1. The proposed method may require training the model for multiple rounds, which may not scale up to datasets with millions of examples.
2. The proposed method introduce several additional hyperparameters such as the K in k-means clustering, the training epochs, but the effect of these components are not studied.
3. The connection between the theory and the proposed method is not clear. More explanation on this matter would be appreciated.

**Questions:**

See above

**Limitations:**

Yes

---

> ### Author Rebuttal · Authors · 2024-08-07
>
> Thank you for your feedback. We appreciate your positive comments on the motivation, clarity, and extensive experiments of our study. We address your concerns and provide clarifications below:
>
> #### **1. Scalability Concerns:**
> The S2L method requires only one additional round of 3-epoch training on a smaller model (e.g., Pythia-70M) to select a subset of training data. Training the 70M proxy on the MathInstruct dataset with 262K examples takes only 30 minutes with a batch size of 128. During this training, the loss trajectory of each example is saved, which is then used to cluster and select representative data. This selected subset can be reused for multiple fine-tuning sessions on larger models, significantly reducing computational costs compared to multiple rounds of full-scale training. For example, training the 7B model (Llama-2-7B in Table 2) with full data for *1 epoch* took 10 hours with a batch size of 64 and a maximum sequence length of 512, consuming 48GB of memory per NVIDIA A6000 GPU with 8 GPUs. As shown in Table 2, with the cost of 30 minutes of training for a small 70M reference model, we can reduce the training time of the target 7B model by half without losing performance. The cost of full-scale training goes up significantly when scaling to millions of examples, making S2L a more efficient and scalable approach.
>
> **New results:** To further improve scalability, we conducted experiments by training the proxy on a smaller sample of the data (100K examples) for the same number of epochs (3 epochs) and saving the loss for all examples. Training the proxy on this smaller sample reduces the training time to approximately 12 minutes while maintaining performance, demonstrating S2L’s efficiency and scalability for large datasets. Please refer to Figure 1 in the one-page PDF attached in the global rebuttal ([link](https://openreview.net/attachment?id=lyJkoUGNPM&name=pdf)) for detailed per-dataset and average accuracy results. We'll add the results to the paper.
>
> #### **2. Hyperparameter Analysis:**
> **New results:** We conducted a detailed analysis on the effect of K (number of clusters). **The findings summarized in Figure 2 of the one-page PDF attached in the global rebuttal ([link](https://openreview.net/attachment?id=lyJkoUGNPM&name=pdf)) demonstrate that S2L maintains high performance across different values of K,** indicating the method is not sensitive to the choice of K. Based on our analysis, we chose K=100 for our experiments because it provided the best average accuracy over the math evaluation datasets and use it for the medical dataset without further tuning, which confirms the robustness/stability of our method.
>
> #### **3. Theory and Method Connection:**
> The theoretical foundation of S2L is based on the observation that, with a small curvature (which is typically the case during fine-tuning), examples with similar loss trajectories have similar gradients during training. Based on this observation, we cluster the examples based on their loss trajectories and then randomly sample examples from each cluster. This approach ensures that the subset's gradient captures the full data's gradient during training.  Because the gradients of the subset and the full data are similar, each gradient update on the subset closely approximates an update on the full data. This similarity guarantees that training on the subset using (incremental) gradient descent maintains similar training dynamics and converges to a solution comparable to training on the full dataset. Our empirical results show that sampling an equal number of examples from different loss-trajectory clusters improves performance by reducing redundancy in the data. This method ensures that the selected data is high-quality and representative, enhancing the overall efficiency and effectiveness of the training process.
>
> We hope the additional explanations and analyses address your concerns. Thank you once again for your valuable feedback.

---

> ### Author Response · Authors · 2024-08-12
>
> Dear Reviewer mU1m,
>
> We hope our recent rebuttal has been helpful in addressing your concerns. If you have any remaining questions, please don't hesitate to let us know—we’d be more than happy to discuss further. We truly appreciate the time and effort you’ve put into reviewing our work, so we want to ensure we’ve adequately responded to your feedback.
>
> Best regards,
>
> SmallToLarge (S2L) Authors

---

### Author Rebuttal · Authors · 2024-08-07

Thank you for your valuable feedback and for recognizing the strengths of our work, including its clear motivation, extensive experiments, and unique contributions to enhancing data efficiency for large language models (LLMs). We have conducted additional experiments and analyses to address the concerns raised and provide new insights, summarized in the attached one-page PDF.

**Figure 1. Scalability:**
To further demonstrate the scalability of the proposed S2L method, we conducted experiments by training the proxy on a smaller sample of the data (100K examples) for the same number of epochs (3 epochs) and saving the loss for all examples. The results, shown in Figure 1, confirm that S2L remains effective when the proxy model is trained on a smaller subset of training data and therefore is scalable to larger datasets without a proportional increase in computational costs.

**Figure 2. Robustness to Clustering Parameter (K):**
We conducted detailed experiments varying the clustering parameter K, as shown in Figure 2. The results demonstrate that S2L maintains high performance across different values of K, highlighting the robustness of our method to different clustering parameter choices. We chose K=100 for our experiments as it provided the best average accuracy across the evaluation datasets for the math reasoning task.

**Figure 3 & 4. Convergence and Data Efficiency:**
Figure 3 extends the analysis presented in the main submission, showing the relative accuracy to full data as the data size increases, with consistent total training iterations/steps for all results. Both in-domain and overall average accuracy are shown. S2L consistently outperforms other methods, such as Random, High Learnability, and Facility Locations, which were the best-performing baselines based on the results we presented in our paper, in terms of relative accuracy to full data. This underscores the efficiency and effectiveness of S2L in achieving comparable or superior performance with fewer data and fewer training iterations.

Figure 4 in the one-page PDF illustrates the relative accuracy to full data across different epochs, comparing S2L-selected data and full data with the same number of epochs. Both in-domain and overall average accuracy are shown. S2L demonstrates superior performance with fewer data and fewer training iterations.

**Figure 5. Proxy Models:**
Reviewers also expressed interest in understanding whether different small models could serve as effective proxies. We used GPT-2 (124M) and Pythia-160M as proxy models for data selection to train Pythia-410M. The results, illustrated in Figure 5, show that both proxy models perform comparably in guiding the data selection, demonstrating the versatility and effectiveness of using different small models for S2L.

We hope these additional explanations and analyses address your concerns and demonstrate the robustness, scalability, and effectiveness of the S2L method. Thank you once again for your valuable feedback.

---

### Decision · Program_Chairs · 2024-09-25

**Decision:**

Accept (poster)

**Comment:**

This paper provides a method for dataset selection for fine-tuning small LMs by looking at the the training trajectory. The experimental evidence is quite compelling (especially the extra results show in the rebuttal) which shows that gains from their method seem to improve/maintain with scale. While it would be interesting to see this method applied to pretraining dataset selection, but I understand this is out of scope for this work. The reviewers agree that this method is novel and interesting, and the experimental evidence is compelling, additionally the most negative review seems to ask for baselines that are known to not work well in the literature. For these reasons I vote to accept the paper.